# *Burkholderia pseudomallei* Complex Subunit and Glycoconjugate Vaccines and Their Potential to Elicit Cross-Protection to *Burkholderia cepacia* Complex

**DOI:** 10.3390/vaccines12030313

**Published:** 2024-03-15

**Authors:** Alexander J. Badten, Alfredo G. Torres

**Affiliations:** 1Department of Microbiology & Immunology, University of Texas Medical Branch, Galveston, TX 77555, USA; ajbadten@utmb.edu; 2Institute for Translational Sciences, University of Texas Medical Branch, Galveston, TX 77555, USA; 3Department of Pathology, University of Texas Medical Branch, Galveston, TX 77555, USA

**Keywords:** *Burkholderia pseudomallei*, *Burkholderia mallei*, *Burkholderia cepacia* complex, melioidosis, glanders, cystic fibrosis, vaccines, subunit, glycoconjugate

## Abstract

*Burkholderia* are a group of Gram-negative bacteria that can cause a variety of diseases in at-risk populations. *B. pseudomallei* and *B. mallei*, the etiological agents of melioidosis and glanders, respectively, are the two clinically relevant members of the *B. pseudomallei* complex (Bpc). The development of vaccines against Bpc species has been accelerated in recent years, resulting in numerous promising subunits and glycoconjugate vaccines incorporating a variety of antigens. However, a second group of pathogenic *Burkholderia* species exists known as the *Burkholderia cepacia* complex (Bcc), a group of opportunistic bacteria which tend to affect individuals with weakened immunity or cystic fibrosis. To date, there have been few attempts to develop vaccines to Bcc species. Therefore, the primary goal of this review is to provide a broad overview of the various subunit antigens that have been tested in Bpc species, their protective efficacy, study limitations, and known or suspected mechanisms of protection. Then, we assess the reviewed Bpc antigens for their amino acid sequence conservation to homologous proteins found in Bcc species. We propose that protective Bpc antigens with a high degree of Bpc-to-Bcc sequence conservation could serve as components of a pan-*Burkholderia* vaccine capable of protecting against both disease-causing groups.

## 1. Introduction

*Burkholderia* are a ubiquitous genus of Gram-negative bacteria primarily consisting of saprophytes and rhizobacteria [1]. Given their specialization for this environmental niche, most of the 120-or-so species of *Burkholderia* are relatively innocuous to humans and are not known to cause overt disease [2,3]. However, there are two prominent disease-causing groups: the *B. pseudomallei* complex (Bpc) and the *B. cepacia* complex (Bcc). The former notably contains *B. pseudomallei* and *B. mallei*, the causative agents of melioidosis and glanders, respectively, which are designated as tier 1 select agents due to their status as potential biothreat agents [4]. Moreover, *B. pseudomallei* is linked to substantial global disease incidence and mortality rates [5], establishing it as the *Burkholderia* species of utmost concern from a public health and safety perspective. While Bcc species are neither biothreat agents, nor are they associated with particularly high incidence rates, they can still significantly affect the quality of life of at-risk individuals, and can even lead to death [6,7,8].

*B. pseudomallei* is found in tropical and subtropical climates where soil conditions are conducive for growth [9], with Southeast Asia and Northern Australia being considered the major hotspots of the disease [5]. However, a comprehensive modeling study suggests that disease incidence is significantly underreported in regions of South and Central America, the Middle East, and Africa [5]. Others have predicted that global climate change could lead to an increase in the incidence rate of melioidosis and could allow the bacteria to spread to areas where it was previously not considered endemic [10,11,12]. This latter prediction has materialized into reality in recent years, with *B. pseudomallei* being isolated for the first time from environmental samples in Mississippi, USA [13], and Southeast Queensland, Australia [14], suggesting a spread of the bacteria to new areas of endemicity. In terms of disease presentation, melioidosis can have a particularly wide array of symptoms affecting various bodily systems [15], leading some to refer to the pathogen as “the great mimicker” [16,17]. In general, roughly 9 out of 10 melioidosis patients experience an acute form of disease, the most common manifestation of which is a pneumonia-like illness that often proceeds to bacteremia [15,18]. The chronic form of disease generally presents as a more tuberculosis-like illness, and is often associated with development of ulcers and abscesses at various sites of the body [15,18]. While chronic infection has a relatively low mortality rate, the acute form of disease is lethal in roughly 10% of cases under ideal treatment conditions [15,18]. In regions where reliable, accurate diagnosis cannot be achieved or proper antibiotics cannot be prescribed, mortality to acute infection increases to over 40% [15,18]. Taking into consideration the underreporting of cases, melioidosis is estimated to result in between 36,000 and 227,000 deaths annually across the globe [5].

*B. mallei* represents a clonal derivative of *B. pseudomallei* that has undergone successive rounds of reductive genome evolution, allowing it to become an obligate mammalian pathogen whose primary reservoir is solipeds [19]. Thanks to stringent global culling and monitoring practices, glanders has been eradicated in most developed countries, though sporadic outbreaks occur in other parts of the world [20,21]. Thus, human cases of glanders are rare, and are usually only observed in lab workers and in people who work closely with horses in regions of the world where the disease is still endemic [22]. Despite the low disease incidence, quite a few *B. mallei* vaccines have been tested in animal models because *B. mallei*, like *B. pseudomallei*, is considered a tier 1 select agent, and there is therefore a level of vaccine development support for national defense purposes. Additionally, it should be noted that nearly all functional *B. mallei* genes are identical or near identical in translated amino acid sequence to their direct homologs in *B. pseudomallei* [19,23]. Thus, eliciting cross-reactivity between *B. pseudomallei* and *B. mallei* using a single vaccine is likely feasible. For this reason, many of the subunit and glycoconjugate vaccines that have been described for *B. mallei* have also been tested against *B. pseudomallei*.

*B. mallei* and *B. pseudomallei* are generally thought to have similar pathogeneses despite their distinct reservoirs. Both species can infect humans via the percutaneous route, inhalation, or ingestion [15,22,24]. To achieve protection against these different routes of infection, *Burkholderia* vaccine researchers have had to adopt new vaccination strategies that induce robust mucosal immune responses characterized by high IgA titers, the presence of Th17 cells, and the presence of antigen-specific lymphocytes in the mucosa-associated lymphoid tissue of the lungs [25]. Upon first entering the body through one of the routes, the bacteria enter the extracellular phase of infection. During this relatively short window, antibodies are likely able promote the clearance of the bacteria via opsonophagocytosis and other Fc-mediated mechanisms [26,27,28,29,30,31,32]. However, when pre-existing antibodies and/or innate immunity are not sufficient to halt the bacteria at the extracellular phase, they can proceed to infect a wide array of phagocytic and non-phagocytic cells, thereby entering the intracellular phase of infection [15,24]. At this stage, antibodies are generally not thought to play a significant role in clearing the bacteria, and, instead, T cell responses are regarded as the primary mediators of bacterial clearance, particularly cytotoxic T cells and Th1 cells [33,34]. Thus, both the humoral and cellular arms of the adaptive immune system are regarded as being critically important for conferring protection to the early and late stages of infection, respectively.

The Bcc group consists of at least 26 species [35], only a fraction of which are associated with disease. Unlike members of the Bpc, Bcc species are ubiquitously distributed across the globe and thus pose a risk to susceptible patients everywhere. Members of the Bcc are consistently identified as one of the most common contaminants of pharmaceutical products, resulting in numerous product recalls and sporadic nosocomial outbreaks [36,37]. As such, individuals with weakened immune systems who are exposed to contaminated material can become sick with a range of disease manifestations largely depending on the route of entry into the body. Cystic fibrosis (CF) patients represent a particularly relevant risk group, as the unique environment of their lungs is conducive to chronic Bcc colonization [7,8]. Affected CF patients can be asymptomatic, though many experience an accelerated decline in lung function. A small subset of patients will develop a rapid-onset necrotizing pneumonia known as Cepacia syndrome, which almost always results in death [6,7,8,38]. Bcc lung colonization is noteworthy for being the only *Burkholderia*-associated disease known to be transmissible from person-to-person, albeit only amongst CF patients [39]. As a result, Bcc-positive CF patients are requested to isolate themselves from other CF patients [37]. Bcc colonization is remarkably difficult to clear with antibiotics, which can cause affected individuals to be given lower priority for potentially life-saving lung transplants [40,41]. Previously, *B. cenocepacia* was considered the most common cause of CF lung colonization, though *B. multivorans* has become the dominant species for the past couple of decades [42,43]. Despite this, *B. cenocepacia* is still quite common and is associated with the highest degree of disease mortality [38]. Other clinically relevant species in the Bcc group include *B. ambifaria*, *B. cepacia*, *B. contaminans*, *B. dolosa*, and *B. vietnamensis* [37]. Only a very small number of vaccination studies have been conducted against Bcc species, which can be at least partially explained by the lack of animal models that effectively recapitulate the human disease [44]. Additionally, the correlates of protection for Bcc colonization of the CF lung are not well-defined, and it is particularly unclear to what extent Bcc-specific adaptive immune responses (i.e., antibodies and T cells) are capable of effectively preventing infection in the unique environment of the CF lung. Finally, because it is primarily CF patients who stand to benefit from a prophylactic Bcc vaccine, there is very little interest at the industry level in pushing a Bcc-specific vaccine through human clinical trials due to the relatively small patient population.

Therefore, some researchers have proposed repurposing Bpc-specific vaccines to treat Bcc infections as a feasible and economically viable solution [45,46,47]. One study recently broke down the cost effectiveness of developing and testing a melioidosis-specific vaccine, ultimately concluding it to be a cost-effective strategy of reducing disease incidence [48]. By extension, a melioidosis vaccine that could additionally protect CF patients from Bcc infection could only provide more incentive for industry-supported vaccine development.

Taking an interest in this concept of a pan-*Burkholderia* vaccine, we decided to take a closer look at which, if any, of the proteinaceous *B. pseudomallei* and *B. mallei* vaccine antigens in the literature may potentially be cross-protective to Bcc species based on amino acid sequence conservation. To assess this, we first downloaded the annotated RefSeq proteome files from the NIH National Center for Biotechnology Information assembly database for all *B. pseudomallei* (Taxonomy ID 28450), *B. mallei* (TaxID 13373), and *B. cepacia* complex (TaxID 87882) strains/isolates that were not flagged as anomalous and that had an assembly level of “complete genome” or “chromosome.” We manually removed some of the resulting files if they appeared to be duplicates of the same strain/isolate. This resulted in a dataset consisting of 147 *B. pseudomallei*, 30 *B. mallei*, and 183 Bcc strains/isolates. A breakdown of the Bcc isolates in terms of their geographic origin and isolation source is provided in Figure 1. It is worth noting that there is some geographic and species bias in the Bcc dataset, as nearly half the proteomes represent *B. multivorans* isolates and nearly half of all isolates were sampled in Europe. By comparison, there is very little representation in Africa and South America due to a lack of high-quality sequences from these regions (Figure 1). The amino acid sequence FASTA files for each previously described *Burkholderia* vaccine antigen were downloaded from UniProt to serve as reference sequences. Such sequences were always taken from *B. pseudomallei* strain K96243 to ensure consistency in our comparisons and because it is one of the most used strains in *B. pseudomallei* vaccine studies. These FASTA files were individually compared to the downloaded RefSeq proteome files using the blastp function of the BLAST+ executables suite v2.14.0 downloaded from the NCBI website [49,50,51]. When running this command, the -e value parameter was set to 0.00005 and the -max_target_seqs parameter was set to 1. For each individual antigen, an output for each strain/isolate comparison was created, and these results were compiled into one dataset using R v4.3.1 [52]. These comparisons’ results are reported in Table 1, Table 2 and Table 3, which are referenced throughout the review as we discuss individual antigens. To keep these tables to a manageable size, we only included the average sequence conservation of the Bcc protein homologs compared to the reference *B. pseudomallei* strain K96243 sequences. Unless stated otherwise in the review, all antigens described had > 90% sequence coverage, > 99% sequence similarity, and were present in > 90% of the annotated proteomes analyzed when comparing sequence conservation across *B. pseudomallei* and *B. mallei* strains/isolates.

## 2. Flagellar Proteins

*B. pseudomallei* and members of the Bcc group are motile due to the presence of one or more flagella. *B. mallei* is notably non-motile, having lost the ability to express a functional flagellum likely due to the presence of an insertion sequence in the *fliP* gene [53]. Flagella have been implicated as potential virulence factors in mouse models of melioidosis [54,55] and in Bcc colonization of the lung [56,57]. They are also potent mediators of inflammation due to their TLR5 and NLRC4 inflammasome agonistic activity [57,58,59,60,61]. The flagellum consists of three major domains: a basal body, a hook, and a long filament that spans into the extracellular milieu. The transmembrane basal body component serves as a motor, utilizing the electrochemical gradient of ions across the bacterial membrane to generate thrust. The hook sits on the surface of the bacterial outer membrane and acts as a flexible linker that transmits torque from the basal body to the filament. Lastly, the filament is the largest and most visible component of the flagellum, consisting of thousands of repeating flagellin (FliC) subunits that serve as a sort of molecular propeller [62]. FliC represents the first proteinaceous antigen characterized for *Burkholderia*, and a handful of studies have explored it as a vaccine antigen either alone or as a carrier protein in lipopolysaccharide (LPS) glycoconjugate vaccines [63,64,65,66,67,68]. Our lab has also evaluated the hook-associated protein FlgL as a candidate vaccine antigen [69,70,71].

### 2.1. FliC (BPSL3319)

FliC has numerous properties that make it an ideal target for humoral immunity. The relatively high level of expression and repetitive pattern of polymerized FliC subunits likely allows them to efficiently cross-link B cell receptors, leading to more efficient B cell activation [72]. Additionally, the flagellar filament is localized deep in the extracellular environment, far away from the bacterial glycocalyx which could otherwise interfere with B cell receptor and antibody binding [72]. FliC is also the component of flagella that activates TLR5 and the NLRC4 inflammasome, meaning that FliC has self-adjuvanting properties when used in a subunit vaccine [58,59,60,61]. Unsurprisingly, given these favorable B cell stimulatory qualities, FliC-specific antibody titers are an established serodiagnostic marker of melioidosis, though such titers do not appear to correlate with improved disease outcome [73,74,75,76,77]. The FliC of both *B. pseudomallei* and *B. cenocepacia* has additionally been found to contain post-translational modifications (PTMs) in the form of species- and strain-specific patterns of glycosylation [78,79]. In *B. cenocepacia*, these glycans appear to partially block the induction of TLR5 and serve to reduce FliC’s immunogenicity [79]. It is currently unclear how these glycans affect antigenicity, either by blocking underlying B cell epitopes or by serving as B cell epitopes themselves.

Brett et al. first isolated FliC from *B. pseudomallei* strain 319a and demonstrated its antigenic potential [63]. This seminal study determined that diabetic rats passively immunized with polyclonal antibodies specific to FliC were partially or fully protected from i.p.-delivered *B. pseudomallei* strain 316c, with survival rates ranging from 20 to 100% depending on the inoculum size and route of antibody delivery. It was further demonstrated in vitro that these antibodies could inhibit the bacteria’s motility, suggesting a possible mechanism of action [63]. Subsequent studies on this mechanism are contentious, with an early study by DeShazer et al. indicating that flagella are dispensable in Syrian hamster and diabetic rat i.p. models of infection, implying that flagella-mediated motility is not necessary [80]. However, Chua et al. found that a flagellin knockout *B. pseudomallei* strain KWH was significantly attenuated in both i.p. and i.n. infection routes using BALB/c mice, suggesting different virulence mechanisms in these rodent models [54]. Another study was then unable to reproduce these results, finding that flagella was essentially dispensable in BALB/c mouse models of infection [81]. Regardless of the exact mechanism, whether FliC-specific antibodies are mediating protection by inhibiting bacterial motility or through Fc-mediated mechanisms of immunity, this initial passive transfer study by Brett et al. clearly demonstrated that FliC-specific antibodies provide some level of protection against *B. pseudomallei*, at least in the diabetic rat infection model [63].

Another group later created a DNA vaccine encoding the *B. pseudomallei fliC* gene [65,66]. The authors of these studies immunized female BALB/c mice intramuscularly with the *fliC*-encoding plasmid or subcutaneously with recombinant FliC in Freund’s adjuvant. An intravenous challenge with 10^5^ CFU *B. pseudomallei* was administered, resulting in 83% protection in the DNA-vaccinated mice, 50% in mice given the recombinant protein, and 0% in the control mice. While statistical significance was not reported for this study, the DNA vaccine appeared to be the superior vaccine formulation, despite eliciting lower overall IgG titers than the recombinant protein. Interestingly, the animals receiving the DNA vaccine had relatively balanced IgG1/IgG2a titers, whereas the animals receiving the recombinant protein were markedly more IgG1-skewed, suggesting Th2 humoral-biased immunity. Furthermore, splenocytes from the DNA vaccinated group exhibited a significantly higher propensity to proliferate and secrete IFN-γ upon ex vivo stimulation with FliC compared to splenocytes recovered from mice immunized with the protein, providing additional support that Th1 responses were stronger in animals receiving the DNA vaccine [65]. These results implicate T cell responses as playing a significant role in protecting these mice from lethal challenge. The group then added immunostimulatory CpG motifs to their DNA vaccine to skew the immune response further towards a Th1 phenotype, though they did not report mouse survival data for these experiments. Instead, they demonstrated that the inclusion of these CpG motifs led to a statistically significant decrease in liver but not spleen colonization using the same intravenous challenge model [66]. Providing additional support that FliC acts as a potent T cell antigen, this protein has also been repeatedly used as the carrier protein in glycoconjugate vaccines to *Burkholderia* [64,67,68]. Such vaccines have exhibited a strong ability to generate high antigen-specific IgG titers to *Burkholderia* surface polysaccharides, suggesting that T-cell-dependent activation of these B cell subsets has been achieved [64,67,68]. However, when compared side-by-side to other potential glycoconjugate carrier proteins, FliC performed worse than Hcp1 and similarly to TetHc in its ability to induce LPS-specific antibody titers, suggesting that it may not be particularly special in its T cell antigenic properties [68]. Further exploration of this topic is needed before claims can be made about FliC-specific T cells and their role in protection against melioidosis.

These early studies that set out to explore a protective role of FliC in vaccination against *B. pseudomallei* used what are now considered outdated challenge models unlikely to recapitulate disease in humans, namely intraperitoneal challenge of diabetic rats [63] and intravenous challenge of mice [65,66]. The more recently explored FliC-incorporating glycoconjugate vaccines have utilized the current “gold standard” model of respiratory-acquired glanders, though similar reports are surprisingly lacking for respiratory-acquired melioidosis. In these glanders studies, there was little evidence to suggest that FliC-specific immunity played any meaningful role in protection, as seen by the similar levels of organ colonization and survival in the animals immunized with FliC-LPS versus those immunized with TetHc-LPS, as TetHc is a non-*Burkholderia* protein. This finding is somewhat unsurprising, given that *B. mallei* lacks an assembled flagellum [67,68]. As there are no peer-reviewed examples of FliC being used to protect against respiratory-acquired *B. pseudomallei*, definitive conclusions cannot be drawn as to FliC’s potential as a melioidosis vaccine antigen. That said, another study discussed unpublished data suggesting that FliC is not protective in such respiratory models, casting some doubt on the antigen’s usefulness as a melioidosis vaccine antigen [82]. It is worth highlighting that Brett et al.’s initial characterization of FliC is the only published study that used *B. pseudomallei*-derived FliC [63], which would have native glycosylation patterns. All subsequent studies have used host-cell translated FliC [65,66] or *E. coli*-expressed FliC [67,68], neither of which would necessarily be expected to retain native-like glycosylation patterns. The extent to which this plays a role in humoral immunity is entirely unexplored and is worthy of future consideration.

Even if FliC does have potential as a melioidosis vaccine antigen, it is significantly hampered compared to most other antigens discussed in this review because it is unlikely to elicit cross-protection to glanders. Similarly, FliC is expected to have only moderate applicability as a Bcc cross-reactive antigen. This is because Bcc species are known to express at least two evolutionarily distinct FliC variants that vary significantly in size and sequence [83]. However, we did find in our bioinformatic screen that *B. cepacia*, *B. cenocepacia*, and most strains of *B. multivorans*, which represent three of the most clinically relevant species of Bcc, express a FliC variant that is similar to that of *B. pseudomallei*. The amino acid sequences of these Bcc FliC variants are 74.0% identical and 84.6% similar to *B. pseudomallei* strain K96243 FliC, suggesting a moderate degree of conservation compared to the other antigens discussed herein (Table 1). Supporting this finding, Musson et al. generated T cell hybridomas specific for a *B. pseudomallei* FliC epitope which cross-reacted strongly with peptides derived from *B. cepacia*-, *B. cenocepacia*-, and *B. multivorans*-derived FliC, suggesting that T-cell-mediated cross-protection could be elicited by this protein [84]. We found in our BLAST analysis that the variants of FliC found in other Bcc species appear to exhibit a generally lower degree of sequence conservation to the N- and C-terminal ends (residues 1–160 and 286–388) of *B. pseudomallei* FliC, though the middle region (residues 161–285) lacks any reported sequence conservation. It is therefore questionable as to whether FliC could elicit cross-reactive immunity to other Bcc species. In summary, FliC’s role as a protective antigen in respiratory-acquired melioidosis is contentious, it is unlikely to protect against glanders to any meaningful capacity, and the extent of cross-reactivity to Bcc FliC is likely to be highly species-dependent, but theoretically possible. Further investigation is needed to confirm these assertions.

### 2.2. FlgL (BPSL0281)

Our laboratory identified the hook protein FlgL in a reverse vaccinology approach designed to screen for potentially antigenic surface proteins [69]. This protein has been included in three separate studies in which different carrier proteins were tested in gold nanoparticle vectored LPS glycoconjugate vaccines to melioidosis [69,71] and glanders [70]. In general, the results when using FlgL have been somewhat mixed. In the first study, it was found that no difference was observed between the carrier proteins in their ability to induce LPS-specific IgG when administered intranasally [69]. Despite this, the construct containing FlgL as the carrier protein outperformed every other construct in its ability to protect mice from a lethal intranasal challenge of *B. pseudomallei* strain K96243, with 90% of mice surviving the challenge compared to 10–20% in the groups receiving LPS conjugated to Hcp1, BpaE, or bovine serum albumin (BSA) [69]. This suggested that FlgL-specific immunity was playing an important role in protecting these mice. However, in a follow-up experiment, we tested a wider array of carrier proteins, and the FlgL construct was not similarly protective [71]. One potential explanation of this discrepancy could be the different adjuvant systems used in these separate studies. Somewhat unsurprisingly, the FlgL-LPS construct was also not protective against a lethal intranasal challenge of glanders, which can likely be attributed to the lack of flagellum assembly in this species [70]. Further studies are needed to confirm whether FlgL is protective against melioidosis. If it is, future researchers may explore whether protection is mediated primarily by FlgL-specific antibodies or T cells, as FlgL is likely much less accessible for antibody binding than FliC. Further exploration of FlgL may be worthwhile given FlgL’s higher degree of Bpm-to-Bcc conservation compared to FliC (Table 1).

**Table 1 vaccines-12-00313-t001:** Bcc-to-*B. pseudomallei* amino acid sequence conservation of antigens predicted or known to be accessible for antibody binding.

*B. pseudomallei* K96243 Locus	Protein Name	Antigen Class	Absent in *B. mallei*?	Bcc % Identity to K96243	Bcc % Similarity to K96243	Bcc % Coverage to K96243	% of Bcc Strains with Gene
BPSL2151	BamA	β-barrel		94.0	96.9	99.8	100
BPSL0999	Omp1	Lipoprotein		93.5	95.9	99.2	100
BPSL2765	Omp7	Lipoprotein		85.5	93.1	100	100
BPSL2522	Omp3	Other		91.6	92.9	99.7	100
BPSL1552	OmpW1	β-barrel		78.3	88.0	98.8	100
BPSS0879	OpcP	β-barrel		75.4	82.4	100	100
BPSL0281	FlgL	Flagella	*	67.9	80.8	100	100
BPSL1972	Bucl8	β-barrel		78.0 ‡	88.2 ‡	98.1 ‡	99
BPSS0708	OpcP1	β-barrel	Partially	75.7	85.5	99.9	98
BPSL3319	FliC	Flagella	*	74.0 †	84.6 †	98.2 †	59 †
BPSL2704	OmpW2	β-barrel		87.8	94.1	91.0	56
BPSS1593	PilV	Pilin	Partially	42.8	56.0	78.3	25
BPSS1532	BipB	T3SS		33.2	53.2	55.6	5
BPSS1993	MprA	Other	Absent	83.3	89.6	99.3	4
BPSS1529	BipD	T3SS		32.2	55.1	58.0	1
BPSS1531	BipC	T3SS		0	0	0	0

* *B. mallei* does not express any flagella, but retains these genes. ‡ Only compared residues 1–495. C-terminal collagen-like domain was excluded. † Only Bcc FliC homologs with full gene coverage (>95%) were included in these calculations.

## 3. Type 3 Secretion Systems

Type 3 secretion systems (T3SS), also known as injectosomes, are syringe-like structures capable of delivering substrates across the plasma membrane of nearby cells or across the endosomal membrane of infected eukaryotic cells [85,86]. This largely implicates them as virulence factors in a wide variety of Gram-negative bacteria including *Yersinia pestis*, pathogenic *E. coli*, *Salmonella enterica*, *Shigella*, and *Pseudomonas aeruginosa*. While most of the T3SS components are localized within the cell envelope where they cannot be readily recognized by antibodies, there are at least two components that are surface-exposed. These include a needle-like complex that extends from the bacterial surface towards the target cell and a hetero-oligometric translocation pore that sits at its tip [85,86]. Some groups have suggested that the outer membrane-embedded ring may be targetable by antibodies as well, though to date, targeting this domain in other bacterial species has had little success [87,88,89,90]. Antibodies towards the translocation pore would be predicted to have the highest likelihood of blocking the activity of the T3SS, with antibodies towards the needle or outer membrane ring being perhaps more likely to elicit Fc-mediated mechanisms of protection.

*B. pseudomallei* expresses two T3SSs belonging to the Hrp-2 group (T3SS-1 and T3SS-2) and one from the Inv/Mxi-Spa group (T3SS-3) [91]. T3SS-2 and -3 are conserved in *B. mallei*, but T3SS-1 is not [91,92]. Bcc T3SSs have minimal overlap with those of the *B. pseudomallei* complex, and there is only one T3SS that is broadly conserved across Bcc species, termed the Bcc injectisome (BCI) [91]. Evolutionary cousins of the T3SS-1, -2, and -3 are found in some Bcc isolates, but only the T3SS-2 is found with any significant degree of sequence conservation in *B. ubonensis* and *B. stagnalis*, two species of little clinical significance [91]. *B. multivorans* and *B. cenocepacia*, the two most clinically relevant Bcc species in CF lung colonization, are reported to only express the BCI [91]. Thus, T3SSs are predicted to have no applicability as Bcc cross-protective antigens.

Despite this, the T3SS-3 is one of the most critical and well-studied virulence determinants in *B. pseudomallei* and *B. mallei*, making it an attractive target for vaccines or therapeutics against melioidosis and glanders. This topic has been extensively reviewed elsewhere [93]. T3SS-3 secreted effectors are required for endosomal escape during the intracellular replication cycle [94], and dissemination of the bacteria from the lungs in intranasally challenged animals [95]. There are seven confirmed T3SS-3 effector proteins including BopA, BopC, BopE, BapA, BapC, BprD, and CHBP [93]. While such effectors are unlikely to play a significant role in humoral immunity given their intracellular localization, they may represent particularly promising T cell antigens. This is because, during the intracellular phase of infection, T3SS substrates are often the first bacterial proteins that make their way into the cytoplasm where they can be processed by the proteasome into MHCI fragments to subsequently stimulate CD8 T cell immunity. This strategy of using T3SS substrates as T cell antigens has been employed with success in *Salmonella enterica* vaccines [96,97] and a similar approach has been utilized with type IV secretion system substrates in *Coxiella burnetii* [98], which are similarly injected into the cytoplasm from within the endosome. However, a key difference in these organisms is that *S. enterica* and *C. burnetii* tend to survive and replicate within the endosome for hours or days, whereas *B. pseudomallei* escapes the endosome within minutes of being internalized [99,100,101,102]. Given this extremely short period of time, it is possible, perhaps even likely, that *Burkholderia* T3SS-3 substrates do not have any advantage as T cell antigens over other proteins. Regardless, based on this information two general strategies for vaccinating against the T3SS-3 have been employed: inducing humoral immunity to the translocation pore [103,104] and inducing T cell immunity towards T3SS-3 effector proteins [82].

### 3.1. T3SS-3 Translocation Pore Proteins

The translocation pore of the T3SS-3 consists of BipB (BPSS1532), BipC (BPSS1531), and BipD (BPSS1529). Stevens et al. first attempted to use BipD as a vaccine antigen, pairing it with Sigma Adjuvant System (SAS) and administering it intraperitoneally in BALB/c mice [103]. The authors ultimately did not observe any protection from a lethal i.p. challenge of *B. pseudomallei* strain K96243, despite robust BipD-specific IgG titers [103]. Druar et al. followed up with a more complete investigation of the T3SS-3 needle tip proteins as antigens, testing BipB, BipC (N-terminal half or C-terminal half), and BipD in Freund’s adjuvant [104]. A single dose of any of these four antigens delivered intraperitoneally did not confer any detectable protection against 30 × LD_50_ of i.p.-delivered *B. pseudomallei* strain Ashdown, again despite robust IgG titers [104]. Curiously, serum from experimentally infected mice did not react to these proteins, while serum from recovered human melioidosis patients did [104]. This could indicate that the lack of protective efficacy in mice may be attributed to the intraperitoneal mouse challenge model employed by the authors. Supportive evidence suggests that the T3SS-3 is expressed at low levels when grown in the lab, but then becomes transcriptionally upregulated when human serum is present, within the environment of the murine lung, or when salt concentrations are high [104,105,106,107]. This may suggest that the amount of T3SS-3 available for antibody binding during the extracellular phase of infection may be dependent on the initial growth conditions of the bacteria as well as its route of entry into the body. Future studies should investigate whether intranasally infected mice develop antibody titers to the needle tip proteins or whether needle tip protein vaccination is protective in intranasal models of infection. Evaluating the needle component, BsaL, as a vaccine antigen may also be worthy of consideration. While BsaL antibodies are less likely to antagonize T3SS-3 activity, they could still prove efficacious through Fc-mediated mechanisms of protection.

### 3.2. T3SS-3 Effector Proteins

BopA (BPSS1524) plays a significant role in intracellular survival, possibly by inhibiting autophagy [93], and is the only T3SS-3 effector that has been tested as a vaccine antigen [82]. In this study, *B. mallei*-derived BopA was adjuvanted with CpG ODN 2395 and immune-stimulating complex AbISCO 100, and intranasally administered into BALB/c mice. The animals were then challenged intranasally with 2 × LD_50_ of *B. mallei* strain ATCC 23344 and were completely protected from mortality over the course of 21 days. By comparison, only one out of eight animals given BSA in the same adjuvant survived past day five [82]. The authors also attempted a heterologous vaccination/challenge model in which the animals were immunized intranasally with the *B. mallei*-derived BopA in cationic liposome-DNA complex adjuvant before being challenged intranasally with 2 × LD_50_ of *B. pseudomallei* strain 1026b. In this model, 60% of the immunized animals survived to the endpoint of the study at day 60, whereas control animals all succumbed within the first week after challenge. Furthermore, the other two antigens tested in this study, BimA and LolC, failed to elicit long-term protection to the same degree [82]. Altogether, this indicates that BopA-specific immunity is protective against respiratory-acquired *B. mallei* and *B. pseudomallei*, likely through T-cell-mediated mechanisms, though the authors did not directly confirm this. Surprisingly, BopA has not appeared in any subsequent vaccine studies despite exhibiting promising results in “gold standard” infection models of melioidosis and glanders. Future studies may try pairing promising B cell antigens, such as LPS or capsular polysaccharide (CPS), with BopA to determine if the two complement each other. T cell responses should also be measured to confirm their role in mediating protection. Furthermore, since BopA showed such promise in this initial study, it may warrant exploring other T3SS-3 effectors as protective antigens.

## 4. Type 5 Secretion Systems

Type 5 secretion systems (T5SS), also known as autotransporters (ATs), are remarkably efficient proteins capable of inserting themselves into the outer membrane to subsequently carry out some effector function in the extracellular space [108,109]. These proteins are involved in many functions, ranging from adhesion, invasion, biofilm formation, cytotoxicity, and immunomodulation. They consist of a translocator domain, which inserts itself into the outer membrane to serve as a pore and a passenger domain, which is extruded across the membrane to carry out the AT’s function [108,109]. Notably, the translocator domain consists of a β-barrel motif. Such motifs make the expression of recombinant proteins in their native conformations particularly challenging, a topic discussed in greater detail later in the section on general β-barrel proteins. As a result, most studies of ATs as subunit vaccine antigens have instead opted to express only the passenger domain of the protein.

There are 11 predicted ATs in *B. pseudomallei* [110]. These can be classified into two distinct groups: classical ATs and trimeric ATs. Nine of the predicted ATs fall into the latter category, including BpaA, BpaB, BpaC, BpaD, BpaE, BpaF, BoaA, BoaB, and BimA [110]. In Gram-negative bacteria, trimeric Ats are almost always involved in adhesion and invasion [110]. BimA is one of the very few exceptions to this rule, whose function is to induce actin polymerization in infected host cells as a form of intracellular motility leading to dissemination of the bacteria into adjacent cells [111]. BimA is thus one of the most well-studied ATs in *B. mallei* and *B. pseudomallei*, and represents a critical virulence factor. The remaining two ATs, BcaA and BatA, are classical ATs which tend to have a more diverse array of possible functions. Our current understanding of these two proteins suggests that the former is involved in host cell invasion [112], while the latter appears to have lipase activity and may be required for efficient intracellular replication and/or survival [113].

ATs in Bcc species are not as well characterized as those in the Bpc, though one study screened eight different species of the Bcc and identified eight evolutionarily distinct lineages of trimeric ATs [114]. We looked further into the *B. cenocepacia* representatives of six of these lineages and found that none had any significant sequence conservation to trimeric ATs of the Bpc, exhibiting poor sequence coverage and low sequence identity (34–57%) to BpaA, BpaC, or BpaE. We were unable to find any information regarding non-trimeric ATs in the Bcc. The extreme variability of the trimeric ATs in Bcc and their apparently poor conservation between the Bpc and Bcc leads us to the conclusion that they would make poor cross-reactive vaccine antigens. That said, AT sequences were largely incompatible with our BLAST analysis pipeline due to the presence of multiple amino acid tandem repeat sequences of highly varying length. As such, we were not able to compare the Bcc AT sequences to those of *B. pseudomallei* strain K96243 using the same pipeline as the other proteins discussed in this review. For consistency, we have opted not to include these comparisons in Table 1. To date, three of the above Bpc ATs have been tested as vaccine antigens to Bpc pathogens, including BimA [82], BpaE [69,70,71], and BatA [115].

### 4.1. BimA (BPSS1492)

BimA is a trimeric AT harboring proline-rich motifs and two regions with significant homology to WASP domain-2. These domains allow the bacteria to recruit host cell actin to its pole [111]. BimA also harbors a variable number of PDASX repeats that are directly involved in inducing actin polymerization, and variants of BimA harboring a higher number of repeats appear to polymerize actin more efficiently [116]. *B. pseudomallei* and *B. mallei* utilize these polymerizing actin filaments to propel themselves across the cytoplasm of the infected cell and to penetrate and infect adjacent cells, thus playing a vital role in dissemination and persistence of the bacteria [117]. While the majority of *B. pseudomallei* isolates appear to express a variant of BimA that is distinct from that of *B. mallei*, a subset of *B. pseudomallei* isolates from Northern Australia, India, and Sri Lanka harbor a *B. mallei*-like variant [118,119,120]. Interestingly, the *B. mallei*-like variant of BimA in melioidosis appears to be associated with increased intracellular persistence [121], increased incidence of neurological symptoms [122], and worse disease outcomes in patients experiencing neurological symptoms [123]. In addition to the two evolutionarily distinct variants of BimA, there is a relatively high amount of variation in the number of PDASX repeats among *B. pseudomallei* isolates, as well as in the number of NIPVPPPMPGGGA repeats found adjacent to the proline-rich domain. Taking this into account, BimA likely represents the protein antigen with the highest degree of *B. pseudomallei* intra-species variability discussed in this review, and is the only antigen that is present in *B. mallei* but lacks significant sequence identity to the *B. pseudomallei* strain K96243 homolog (~71% sequence identity). This represents a potentially significant limitation to vaccinating against BimA, as cross-reactivity between species or even strains may be low compared to the other antigens.

Due to these limitations, only one study has attempted to use BimA as a vaccine antigen. Whitlock et al. expressed BimA from *B. mallei* strain ATCC 23344 without its signal sequence or the C-terminal translocator domain, and formulated it with the adjuvants CpG ODN2395 and immune-stimulating complex AbISCO100. The formulation was administered to BALB/c mice via the i.n. route before challenging via the same route with 2 × LD_50_ *B. mallei* strain ATCC 23344. All vaccinated animals survived to 21 days post-challenge compared to only 12.5% in control animals. The authors performed a follow-up experiment in which they adjuvanted this *B. mallei* variant of BimA with cationic liposome-DNA complex and subsequently challenged animals with 2 × LD_50_ of *B. pseudomallei* strain 1026b. This heterologous challenge model resulted in ~80% protection to the acute phase of infection and ~20% long-term protection. While this was notably better than the control mice, which all succumbed during the acute phase of infection, BimA provided inferior long-term protection compared to vaccination with BopA (60%) [82]. It has not yet been determined if antibodies to BimA can inhibit intracellular motility and/or intercellular dissemination, or if some other mechanism is responsible for the reported protection.

### 4.2. BpaE (BPSS0908)

BpaE (hemagglutinin) is a somewhat typical trimeric AT which plays a role in adherence and invasion [124]. A *B. pseudomallei* BpaE mutant exhibited significantly reduced adherence, invasion, and plaque formation when incubated with A549 cells [124], and exhibited a 40-fold higher median lethal dose in a murine melioidosis infection model [125]. Together, this implicates BpaE as a moderate virulence factor in *B. pseudomallei*. BpaE has also been confirmed as an immunogenic protein during natural infection, as human melioidosis serum and peripheral blood mononuclear cells (PBMCs) are reactive towards it [69,75,125]. BpaE is another one of the proteins identified by our group’s reverse vaccinology screen as a potentially immunogenic surface protein, in which we referred to it as “hemagglutinin” due to its annotation as a hemagglutinin-like protein. The full-length recombinant protein from *B. mallei* strain ATCC 23344 was expressed and chemically conjugated to LPS as a glycoconjugate vaccine [69]. The BpaE-LPS construct consistently elicited high LPS-specific antibody titers [69,70,71]. Despite this, in two separate studies in which respiratory challenges with *B. pseudomallei* were performed, the BpaE-LPS construct failed to elicit meaningful protection [69,70,71]. However, in a glanders infection model, 90% of mice were protected to a low dose challenge of 2 × LD_50_ and 50% of mice were protected from a 25 × higher challenge dose. The control mice in these two experiments exhibited 50% and 0% survival, respectively. In the high dose challenge experiment, we also noted that the animals receiving the BpaE construct appeared to have the highest rate of sterilizing immunity in the lungs, liver, and spleen, though statistical significance could not be established [70]. This may suggest that BpaE-specific immunity is playing a role in limiting dissemination, though, given the scope of this experiment, we could not definitively determine if such protection was due to BpaE-specific immunity or LPS-specific immunity.

### 4.3. BatA (BPSL2237)

BatA, one of the only two known classical ATs in the Bpc, was the final *B. pseudomallei* AT to be functionally characterized from the initial list of 11. In murine models of melioidosis infection, BatA appears to play little to no role in virulence [125]. Furthermore, PBMCs from melioidosis seropositive human donors do not significantly respond to antigen recall with BatA, nor does seropositive donor serum react to recombinant BatA, suggesting that the protein is poorly immunogenic or minimally expressed during *B. pseudomallei* infection [115,125]. In sharp contrast, a BatA knockout strain of *B. mallei* exhibited a reduction in virulence by more than two orders of magnitude in aerosol challenge models in mice, and both horses and mice experimentally challenged with wild type *B. mallei* elicited significant BatA-specific antibody titers [113]. These results may highlight a potential difference in virulence mechanisms between *B. pseudomallei* and *B. mallei*. The authors of the *B. mallei* studies attempted to understand this discrepancy by functionally characterizing BatA in *B. mallei*, finding that it has lipolytic activity and plays a role in intracellular survival and/or replication in J774 murine macrophages [113].

Because of these findings, the *B. mallei* derived BatA passenger domain was subsequently tested by Lafontaine et al. as a recombinant vaccine antigen [115]. The group opted to use a somewhat atypical adjuvant for their initial experiment, that being RECOMBITEK Lyme, a veterinary-approved Lyme disease vaccine primarily consisting of lipidated OspA protein from *Borreliella burgdorferi*. Simultaneous administration of this vaccine via both the subcutaneous and intranasal routes on three separate days resulted in an extremely IgG1-biased immune response indicative of primarily humoral-driven immunity [115]. After three immunizations, the animals were challenged i.n. with 10 × LD_50_ *B. mallei* strain ATCC 23344, resulting in 26% protection to acute infection and 21% protection past two weeks [115]. By comparison, the control animals receiving only the adjuvant were 10% protected from acute infection, and only 5% survived more than 2 weeks [115]. Three of the four surviving vaccinated animals had no detectable bacteria in their lungs, though sterility was not observed in the spleens [115]. Altogether this experiment suggested only lackluster protection afforded by recombinant BatA passenger domain, though it is possible that the lipolytic activity of BatA may have interfered with OspA’s adjuvanting capacity by cleaving its lipid tail, a possibility that was not addressed by the study.

The BatA passenger domain was also cloned into parainfluenza virus 5 (PIV5) to serve as a vector for the antigen [115]. A single dose of this construct resulted in 85% and 74% protection against acute and chronic phases of glanders infection, respectively. Sterility was observed in the lungs of nearly half the surviving mice, but again not in the spleens. By comparison, a PIV5 vector expressing a non-specific tuberculosis protein only conferred 35% and 25% protection against acute and chronic phases, respectively. The authors also explored whether BatA-specific immunity could protect against a heterologous intranasal *B. pseudomallei* strain K96243 challenge. A total of 80% and 60% protection to acute and chronic stages of infection was achieved, respectively, with PIV5-vectored BatA, whereas the PIV5 control vector afforded only 33% and 20% protection. The authors reported almost complete sterility in the lungs of surviving mice, and nearly half the surviving mice lacked bacteria in the spleen. Despite these promising results, the authors reported that PIV5-BatA-vaccinated animals did not elicit significant antibody titers towards recombinant BatA passenger domain, possibly suggesting that humoral immunity is not responsible for this protection [115]. If antibody titers to BatA were indeed low, BatA-specific T cell responses would likely be responsible for the protection reported by the authors. While an IFN-γ ELISpot of BatA-recalled splenocytes resulted in a significant number of spots compared to those from naïve mice, splenocytes from mice vaccinated with the non-specific PIV5 control construct also exhibited a significant number of spots that was not clearly discernible from the PIV5-BatA group. Thus, it is still unclear how the PIV5-BatA vaccine is mediating protection, and evidence for BatA’s use in a melioidosis or glanders vaccine is inconclusive.

## 5. Type 6 Secretion Systems

The type 6 secretion system (T6SS) is sometimes described as a molecular “harpoon” that is most often used by bacteria to deliver harmful effectors to other prokaryotes or eukaryotic cells [126,127]. Less commonly, they can also play a role in scavenging nutrients and metal ions [126,127,128]. The T6SS consists primarily of the harpoon-like structure, a sheath surrounding it, and a membrane-spanning complex and base plate. The shaft of the harpoon consists of polymerized Hcp (TssD) proteins that end in a tip made of VgrG (TssI) and PAAR. Cargo molecules can bind to the shaft or tip for delivery into the recipient cell, though Hcp, VgrG, and PAAR can have effector functions as well [129,130,131]. The sheath consists of polymerized TssB and TssC, the length of which is controlled by TssA at the base. This sheath contracts to thrust the harpoon into an adjacent cell via a contact-dependent manner (virulence) or into the extracellular milieu in a contact-independent manner (nutrient acquisition). Both the sheath and shaft reside primarily in the cytoplasm until the shaft is fired across the cell envelope. Finally, the membrane-spanning complex and base plate consist of TssE-G and TssJ-M, providing a channel for the harpoon-like structure to pass the bacterial envelope. Only TssJ is found in the outer membrane, and it appears to be located exclusively within the lipid bilayer and periplasm [126,127]. In summary, none of the components of contact dependent T6SSs are predicted to be exposed to the extracellular space before or after secretion, indicating that they are likely serving as T cell antigens and not antibody targets when used in vaccination.

The *B. pseudomallei* genome harbors six T6SS loci (T6SS-1-6). There are two distinct numberings of these systems used in the literature, which can be a source of confusion. In this review, we have opted to use the nomenclature introduced by Schell et al. [132]. The T6SS-2 was characterized as a contact-independent system involved in Zn^2+^ and Mn^2+^ acquisition to combat oxidative stress [128]. The T6SS-4 was recently implicated as being involved in biofilm formation, flagella biosynthesis, and intracellular survival in A549 cells, though the exact mechanistic role of the T6SS-4 in these wide-ranging functions has yet to be elucidated [133]. The T6SS-6 is highly conserved across *B. pseudomallei* and all Bcc species, leading Spiewak et al. to conclude that it is the prototypical T6SS of *Burkholderia* [134]. It is currently held that this system is used to impede the growth of other Gram-negative bacteria [134]. Very little is known about the T6SS-3 and -5; however, they, along with the T6SS-2, -4, and -6, do not appear to play any significant role in virulence in mammalian models of infection [134,135,136]. For the sake of developing a cross-protective vaccine, it is worth noting that *B. mallei* has lost the entire T6SS-5 locus as well as seven T6SS-6 genes, likely rendering the T6SS-6 non-functional [132]. Additionally, only the T6SS-6 is broadly conserved across Bcc species, and is thus the only T6SS that could theoretically be targeted for *B. pseudomallei*-to-Bcc cross-protection (Table 2) [134].

The T6SS-1 is the most well-studied T6SS in *B. pseudomallei* and *B. mallei* on account of its clearly defined role in virulence [135,136]. It was first identified in a screen for *B. pseudomallei* proteins that are upregulated during macrophage invasion and intracellular survival [137]. Around the same time, studies in *B. mallei* indicated that overexpression of the VirAG virulence-associated transcriptional regulatory system resulted in upregulation of the T6SS-1 [132]. In the absence of this system, *B. mallei* and *B. pseudomallei*’s ability to replicate in macrophages and induce the formation of multinucleated giant cells (MNGCs), a characteristic disease manifestation, is decreased [136,138]. To date, no cargo molecules have been identified for the T6SS-1. Instead, our current understanding of the T6SS-1’s mechanism of action indicates that the tip protein VgrG [129,130], and potentially PAAR as well [131], are directly responsible for inducing host cell membrane fusion with neighboring cells, leading to the formation of MNGCs.

To date, vaccination towards T6SSs in *B. mallei* and *B. pseudomallei* have primarily focused on the shaft protein of the T6SS-1, Hcp1 (BPSS1498), though the shaft proteins of the T6SS-2-6, as well as the TssM protein from the T66S-1 locus, have also been tested [31,136]. These antigens are not generally thought to be exposed to the extracellular space for recognition by humoral immunity, and would primarily be thought to serve as T cell antigens. Likely due to the generally held belief that both B and T cell immunity are needed for a vaccine to confer durable protection to melioidosis and glanders, most of the studies utilizing Hcp1 have also included a B cell antigen, such as CPS or LPS.

### 5.1. T6SS Shaft Proteins

Multiple studies have demonstrated that Hcp1 is highly immunogenic, reacting particularly strongly to human melioidosis sera [136,139] and glanders sera from mice, horses, and one human [132,140]. PBMCs from melioidosis seropositive donors also respond to stimulation with Hcp1, correlating with patient survival [141]. By comparison, Hcp2 (BPSS0518), Hcp3 (BPSS2098), Hcp4 (BPSS0171), Hcp5 (BPSS0099), and Hcp6 (BPSL3105) are minimally reactive to human melioidosis serum [136], suggesting they may not be similarly immunogenic during an infection. Burtnick et al. screened these Hcp proteins as potential melioidosis vaccine antigens [136]. Female BALB/c mice were intraperitoneally immunized, with one of each of the Hcp proteins formulated in SAS. The animals were subsequently challenged i.p. with ~50,000 CFU of *B. pseudomallei* strain K96243, ~50 times the median lethal dose. In this first trial, Hcp2 exhibited the greatest protective efficacy with 5 of 6 mice surviving to day 42 post-challenge. Hcp1, 3, 4, and 6 all had 2 or 3 animals survive to day 42, while Hcp5 did not confer any protection. However, in a subsequent replicate of this study using only Hcp1, 2, 3, and 6, Hcp6 conferred the highest level of protection with 4 of 12 animals surviving to day 42. In the Hcp1, 2, and 3 groups, only 2, 1, and 3 animals survived, respectively, which was approximately the same degree of protection as afforded by the adjuvant alone group, in which 2 animals survived. This study appears to be the only published use of Hcp2 through Hcp6 in vaccination. Given that Hcp6 provided the highest level of protection in the follow-up experiment, even if only marginally, it may be worth revisiting as a potentially cross-protective antigen to Bcc species due to its extremely high degree of sequence conservation between *B. pseudomallei* and Bcc species (Table 2) [134].

Likely due to the T6SS-1’s clear role in virulence and Hcp1’s potent immunogenicity during natural infection, only Hcp1 has been explored further as a subunit vaccine antigen, despite the somewhat lackluster protection afforded by the protein in the initial screen [136]. In another study, Whitlock et al. tested whether *B. mallei*-derived Hcp1 was protective against a respiratory glanders challenge [82]. When female BALB/c mice were immunized i.n. with the antigen paired with CpG ODN 2395 and immune-stimulating complex AbISCO 100, mice were 80% protected from a 2 × LD_50_ challenge of *B. mallei* strain ATCC 23344, in-line with the other antigens tested in this study (LolC, BimA, BopA) [82]. In this study, Hcp1 conferred markedly higher protection than reported in the Hcp screen [136], though it is unclear if this is due to the different pathogen, immunization/challenge routes, adjuvant systems, or challenge doses.

In recent years, Hcp1 has become a particularly popular component of glycoconjugate vaccines, either directly conjugated to *B. thailandensis*-derived LPS [68,69,70,71] or admixed with CPS that has been conjugated to a non-*Burkholderia* carrier protein [31,142]. Gregory et al. were the first to chemically link LPS to Hcp1, which was in turn linked to a gold nanoparticle vector. When this gold nanoparticle construct was administered in an intranasal immunization/challenge model in mice, it was noted that it elicited markedly higher LPS-specific IgG titers than when FliC or a tetanus toxoid carrier were used as carrier proteins [68]. When mice were subsequently challenged with a low dose of 1.9–2.6 × LD_50_ *B. mallei* strain China7, 89% of the Hcp1-LPS-vaccinated animals survived, which was superior to but not significantly different from animals receiving the tetanus toxoid- or FliC-containing constructs [68]. In a subsequent experiment using a higher dose of 7 × LD_50_ of the same challenge strain, the animals given the Hcp1-LPS formulation were 22% protected, largely in-line with the other two groups [68]. In both studies, the three groups receiving protein-linked LPS exhibited significantly higher protection and lower spleen colonization than the negative control animals. However, the lack of any significant difference in protection between the Hcp1-LPS vaccinated group and those receiving the FliC or tetanus toxoid carriers may indicate that Hcp1-specific immune responses played little to no direct role in the observed protection, which was instead likely mediated primarily by LPS-specific immunity. Regardless, this study at least confirmed that Hcp1 could elicit robust T cell responses, as indirectly indicated by higher LPS-specific IgG titers.

Our group later used the same gold nanoparticle platform to test a wider array of *B. mallei*-derived carrier proteins for their ability to protect against melioidosis and glanders challenges when conjugated to LPS [69,70,71]. In all three of these studies, we did not observe any significant improvement in LPS-specific antibody titers afforded by the inclusion of Hcp1, at least compared to the other carrier proteins that were screened [69,70,71]. On the contrary, it was observed that the Hcp1-containing glycoconjugate elicited the lowest LPS-specific titers in one such study [71]. When a variety of these different constructs were tested for their ability to protect against a respiratory glanders challenge, the Hcp1 glycoconjugate elicited 89% protection, which was more-or-less in line with all the other carrier protein formulations that were assessed [70]. The lack of differences in protection against glanders afforded by the different carrier proteins may again suggest that protection is primarily being mediated by LPS-specific antibodies rather than carrier protein-specific responses. One of the primary arguments for including a strong T cell antigen such as Hcp1 in a *B. mallei* or *B. pseudomallei* vaccine formulation is that it may provide superior sterilizing immunity by providing additional protection from the intracellular phase of infection. However, in the glanders studies, significant differences were not observed in spleen and lung colonization in surviving animals between the different carrier protein groups, and sterilizing immunity was not achieved [70]. When these constructs were used to immunize against a heterologous melioidosis challenge, the Hcp1-containing glycoconjugates failed to elicit any detectable protection [69,71].

Burtnick et al. paired *B. pseudomallei*-derived Hcp1 with a CPS glycoconjugate consisting of CPS chemically linked to the potent T cell antigen CRM197, a non-toxic mutant of diphtheria toxin [31]. The authors demonstrated via IFN-γ ELISpot that PBMCs from mice immunized with Hcp1 or Hcp1 paired with CPS-CRM197 were strongly reactive to stimulation with the protein, further supporting Hcp1’s T cell antigenic properties [31]. When the authors assessed Hcp1’s protective efficacy in vaccination, they found that the combination of Hcp1 and CPS-CRM197 conferred 100% protection against an intranasal challenge of 10 × LD_50_ *B. pseudomallei* strain K96243 compared to 67% in the CPS-CRM197 group, 30% in the Hcp1 group, and 0% in the adjuvant group [31]. This study thus provides some of the strongest support for the inclusion of Hcp1 in a subunit vaccine, demonstrating its protective efficacy with and without the B cell antigen CPS. Most surviving animals in both the combination and CPS-CRM197 groups had sterilizing immunity in the lungs, livers, and spleens, which may indicate that CPS-specific responses were primarily responsible [31]. Based on these results, Biryukov et al. recently performed a similar immunization study using CPS-CRM197 + Hcp1 with or without AhpC, another robust T cell antigen, discussed later [142]. Somewhat counterintuitively, the inclusion of AhpC lowered the degree of protection to an aerosol challenge from 80% to 50%, though both groups performed far better than the adjuvant alone group at 0% survival. Furthermore, bacterial loads in the spleens and lungs of these animals were significantly reduced three days post-challenge compared to controls [142]. When splenocytes from these two immunization groups were recalled with either AhpC or Hcp1, it was determined that AhpC was the more potent T cell antigen, as PBMCs recalled with AhpC elicited 1–2 orders of magnitude higher levels of various Th1- and Th2-associated cytokines compared to stimulation with Hcp1 [142]. Finally, these authors also noted that the Hcp1 + CPS-CRM197 formulation conferred protection to *B. pseudomallei* strain MSHR5855 and a heterologous challenge with *B. mallei* strain FMH, though it is unclear to what extent Hcp1 was responsible for this strain and species cross-protection, given the lack of Hcp1 and CPS-CRM197-alone controls [142].

Another novel approach to Hcp1 vaccination was recently conducted by Zhu et al., who were able to produce *Staphylococcus aureus*-derived membrane vesicles loaded with Hcp1 [143]. This was achieved by fusing the *hcp1* sequence to the *S. aureus* pyruvate dehydrogenase E1 component subunit beta (*pdhB*), a protein known to be efficiently loaded into membrane vesicles. BALB/c mice were immunized with recombinant Hcp1 or the Hcp1-loaded vesicles, either alone or adjuvanted with Freund’s adjuvant. Animals were simultaneously immunized subcutaneously, intramuscularly, and intraperitoneally. Only the animals receiving the adjuvanted vesicles elicited significant Hcp1-specific antibody titers, though this did not translate to any significant difference in protective efficacy between the adjuvanted (70% survival) and unadjuvanted (60%) groups receiving the vesicles after a challenge with 5 × LD_50_ of *B. pseudomallei* strain BPC006. Given that Hcp1 is predicted to mediate protection primarily through T cell responses, it may have been more relevant to perform ELISpots or similar assays using Hcp1-recalled splenocytes from the immunized animals. By comparison, vaccination with recombinant Hcp1 conferred no protection, though this can likely be attributed at least in part to the lack of adjuvant in this group. Finally, five days post-challenge, the adjuvanted and unadjuvanted groups receiving the vesicles had no colonization of the lungs, liver, or spleens [143].

Overall, there appears to be just as much evidence in support of including Hcp1 in a vaccine to *B. mallei* or *B. pseudomallei* as there is weighing against it. The wide range of efficacies reported for this protein could be due to a variety of reasons such as the route of immunization/infection, the adjuvant used, inclusion of other antigens, etc. That said, there is another potential drawback to using Hcp proteins as antigens that may provide additional context for why Hcp1-mediated protection appears so variable. Specifically, it has been demonstrated previously that the Hcp protein of the sci-1 T6SS in enteroaggregative *E. coli* is capable of spontaneously associating into hexameric rings even in the absence of other T6SS components [144]. Furthermore, these hexamers were capable of self-assembling into a shaft-like structure in an unregulated manner such that they could arrange in head-to-head, tail-to-tail, or head-to-tail orientations [144]. This finding has significant implications for using the protein as a subunit vaccine antigen, assuming Hcp1 is similarly capable of this spontaneous assembly. Even if Hcp1 is only being used as a soluble, unmodified antigen, the degree of spontaneous polymerization could result in significant batch-to-batch variability, and, in turn, could significantly impact the pharmacokinetics and pharmacodynamics of the protein. For example, larger particulates are much more readily detected by innate immune cells than non-aggregated antigen due to size-dependent differences in immune recognition [145,146,147]. Furthermore, large particulates have a decreased ability to passively diffuse to secondary lymphoid tissues [145]. In the case of nanoparticle-vectored vaccines such as the gold nanoparticle platform, conjugation efficiency may be directly affected by the degree of Hcp1 polymerization. While these observations are speculative at this point, exploration of this matter appears justified if Hcp1 is to ever make it to regulated pre-clinical or clinical studies, particularly if post-expression modifications are to be utilized.

**Table 2 vaccines-12-00313-t002:** Bcc-to-*B. pseudomallei* amino acid sequence conservation of antigens predicted to serve primarily as T cell targets.

*B. pseudomallei* K96243 Locus	Protein Name	Antigen Class	Absent in *B. mallei*?	Bcc % Identity to K96243	Bcc % Similarity to K96243	Bcc % Coverage to K96243	% of Bcc Strains with Gene
BPSL2096	AhpC	Other		98.9	99.7	100	100
BPSL2287	IscA	Other		95.7	97.2	100	100
BPSL2277	LolC	ABC Transporter		93.3	97.1	100	100
BPSL3105	Hcp6	T6SS	Absent	92.2	97.1	100	100
BPSS0467	PotF	ABC Transporter	Absent	86.1	92.1	99.4	100
BPSS2141	OppA	ABC Transporter		82.8	89.4	96.4	100
BPSL3369	AcoD	Other		74.3	84.7	100	100
BPSL1897	TadE	Other		48.3	60.7	92.6	100
BPSS0171	Hcp4	T6SS		89.2	95.7	100	48
BPSS2098	Hcp3	T6SS		93.1	97.7	100	16
BPSS1498	Hcp1	T6SS		25.6	42.0	97.9	12
BPSS0099	Hcp5	T6SS	Absent	88.6	95.1	100	8
BPSS0518	Hcp2	T6SS	Partially	95.6	98.3	100	2
BPSS1512	TssM	T6SS *		0	0	0	0
BPSS1524	BopA	T3SS		0	0	0	0

* TssM is considered part of the T6SS-1 genomic locus but does not appear to play a functional role in the T6SS-1.

### 5.2. TssM (BPSS1512)

Outside of *Burkholderia*, TssM proteins are generally characterized as important structural components of the membrane-spanning complex of T6SSs. However, this function has not technically been confirmed for the TssM of *B. mallei* and *B. pseudomallei*’s T6SS-1 genomic locus. In fact, knocking out this gene had no effect on intracellular survival or MNGC formation in RAW 264.7 cells [148], suggesting this structural role may not be conserved in the T6SS-1. Instead, most studies of this protein have characterized it as a deubiquitinase that is secreted during the intracellular phase of infection [149]. This protein appears to be directly involved in interfering with NF-kβ and type I IFN responses of infected cells [149], and its secretion is reportedly mediated by the type 2 secretion system [150]. Human melioidosis serum and PBMCs react strongly to TssM, and the latter appears to correlate with survival [75,141]. Burtnick et al. have performed the only vaccination study using TssM, pairing it with CPS-CRM197 [31]. Upon intranasal challenge, the TssM + CPS-CRM197-immunized animals were 80% protected, which was in-line with the group receiving only CPS-CRM197, which were 70% protected. By comparison, the Hcp1 + CPS-CRM197 group was 100% protected, though a statistically significant difference between these groups was not evident. Animals receiving TssM alone were only 20% protected, which was comparable to the animals receiving only Hcp1 [31]. Together, this supports the notion that CPS-specific immunity is primarily responsible for the reported protection, and that the inclusion of TssM appears to be only slightly beneficial.

## 6. Type IV Pili

Type IV pili are long, thin projections that are involved in several functions such as mediating twitching and gliding motility, uptake of DNA, and adherence to eukaryotic cells. These structures primarily consist of repeating subunits known as major pilins, though minor pilins are also present at lower abundance [151]. Like flagellin, polymerized pilin subunits may be particularly antigenic to B cells on account of their repetitive pattern and distance from the bacterial glycocalyx. The *B. pseudomallei* genome harbors eight distinct type IV pilus-associated loci (TFP1-8) that remain only partially characterized [152]. To date, only the minor pilin subunit of TFP7 (BPSS1593; PilV) has been tested in a subunit vaccination approach. While the authors reported that PilV was particularly immunogenic, subcutaneous vaccination with the *B. pseudomallei* strain K96243-derived protein did not protect BALB/c mice from an intraperitoneal challenge of *B. pseudomallei* strain G207 [153]. This failure to confer protection could be due to a multitude of reasons, such as structural differences in the recombinantly expressed protein or low expression of the TFP7 locus in mouse infection models. As it stands, PilV does not appear to be an appropriate subunit vaccine candidate, though the other type IV pilus proteins remain to be characterized.

## 7. General Outer Membrane Proteins: β-Barrels

Not including multi-subunit complexes such as flagella and the various secretion systems, the outer membrane proteome of Gram-negative bacteria largely consists of two major classes of proteins: lipoproteins (discussed in the following section) and proteins containing a β-barrel motif [154,155]. As the name suggests, β-barrels consist of 8–36 antiparallel β-strands that loop back and forth through the outer membrane, arranging into a cylindrical “barrel” shape held together by hydrogen bonds [154]. The core of these proteins is generally hollow, serving as a channel for various cargos (nutrients, waste products, antibiotics, toxins, etc.) in both passive and active transport. The periplasmic and extracellular regions that link the β-strands are often referred to as “loops.” Such loops at the extracellular face are the most likely regions to serve as functional B cell epitopes, given their relative accessibility to the extracellular environment. Conversely, the transmembrane barrel and periplasmic loops would primarily be thought to serve as T cell epitopes.

Because these proteins are embedded in the outer membrane, they consist of a relatively high number of hydrophobic, externally oriented residues, making it difficult to express these recombinant proteins due to solubility issues. Further complicating matters, natively expressed outer membrane β-barrels generally require specialized bacterial machinery, known as the β-barrel-assembly machinery (BAM) complex, as well as a class of chaperones known as holdases to be stably inserted into the outer membrane in their native conformation [154,156]. Structural biologists interested in β-barrels have occasionally been able to circumvent these issues using detergents [157,158], liposomes [159], or lipid nanodiscs [158,160], which mimic the environment created by the bacterial outer membrane and associated holdases. As one might expect from these structural limitations, the study of β-barrel proteins as subunit vaccine antigens is particularly challenging. In the absence of stabilizing agents, protein yields tend to be exceedingly low, the resultant protein is prone to aggregation, and any protein recovered may not be expected to maintain a native-like three-dimensional structure. From an immunological perspective, these last two issues are particularly problematic for B cell responses, as many of the epitopes recognized by B cells are non-contiguous and would not be expected to be retained in aggregated or misfolded protein. Some animal vaccination studies have included detergents to stabilize β-barrel proteins, though many detergents are of questionable biocompatibility and only polysorbate 20 and 80 have been included in FDA-approved vaccines. Furthermore, it is unclear if detergents can retain the protein’s folding conformation in vivo long enough for the antigen to engage with B cell receptors. Despite these limitations, quite a few β-barrel proteins have been explored as subunit vaccine antigens for melioidosis and glanders vaccines.

### 7.1. BamA (BPSL2151)

The first such protein studied as a *Burkholderia* subunit vaccine antigen was an Omp85 family protein (BPSL2151) [27]. Su et al. initially performed a high-throughput screen for human melioidosis serum-reactive peptides derived from *B. pseudomallei* strain D286. They ultimately identified 109 reactive peptides, one of which was derived from BPSL2151 [161]. Protein BLAST of the BPSL2151 sequence largely implicates this Omp85 family protein as being the *Burkholderia* homolog of BamA [162], one of the most highly conserved proteins found in the outer membrane proteome of virtually all known Gram-negative bacteria [163]. This high conservation is due to the integral role of the BAM complex, which serves to mediate insertion of other β-barrel proteins into the outer membrane [163]. Studies in *E. coli* have demonstrated that bacteria are not viable without this system [164], likely due to the vital functions of the myriad β-barrel proteins expressed on the outer membrane. The BamA protein has thus been explored recently as a potential vaccine candidate and target of monoclonal antibodies in *Acinetobacter baumannii* [165,166] and pathogenic *E. coli* [167,168,169].

Su et al. expressed the *B. pseudomallei* strain D286-derived full-length protein in *E. coli*, resulting in inclusion bodies that had to be denatured and refolded [27]. While the protein was not verified to be refolded into a native-like structure, the study demonstrated that human melioidosis serum reacted to the recombinant protein [27]. However, this does not confirm proper folding of the protein, as the reactive antibodies could be limited to linear B cell epitopes on BamA. Given that this protein was first identified by serum reactivity to such linear B cell epitope peptides, melioidosis serum may be expected to react to recombinant BPSL2151, whether it is properly folded or not. Most structural studies of *E. coli* BamA report that detergents or lipid-based carriers are required to maintain the protein in a native-like form. Furthermore, these studies often do not utilize full-length BamA, instead truncating it such that the bulky periplasmic domains are removed [157,159,160,169]. This vaccination study did not report using any such stabilizing agent, nor did they truncate the protein, suggesting the protein may not have been conformationally intact [27].

Regardless, they immunized BALB/c mice intraperitoneally with the recombinant protein and Freund’s adjuvant before challenging intraperitoneally with 10 × LD_50_ of *B. pseudomallei* strain D286. Immunized animals were afforded 70% protection, whereas only 10% of the control animals survived. Spleen, lung, liver, and blood colonization was significantly reduced in the vaccinated group, though sterilizing immunity was not achieved. Finally, the authors provided a potential mechanism of immunity by demonstrating that polyclonal antibodies from the vaccinated animals were capable of opsonizing bacteria and were bactericidal in the presence of complement, suggesting involvement of the classical complement activation pathway [27]. Interestingly, a BamA-specific monoclonal antibody has been described in *E. coli*, which was bactericidal even in the absence of complement. This antibody was capable of directly antagonizing BamA’s ability to mediate the insertion of other β-barrel proteins into the outer membrane, resulting in cell death [168]. We speculate that such direct bactericidal activity in the absence of complement may be achievable using *Burkholderia*-derived BamA if conformational B cell epitopes can be maintained.

Despite these promising initial results and an interest in BamA in other Gram-negative bacteria, *Burkholderia* BamA has received little further interest as a subunit vaccine component. Of all the surface-expressed proteins discussed in this review, BamA exhibits the greatest degree of sequence conservation between the Bpc and Bcc species (Table 1), and thus represents one of the most promising targets for humoral-mediated cross-protection to these pathogens. Specifically, the Bcc homologs of BamA had an average sequence identity and similarity of 94.0% and 97.0%, respectively, compared to *B. pseudomallei*-derived BamA (Table 1). In fact, when limiting this analysis of sequence conservation to only the extracellular loops of BamA, Bcc-to-*B. pseudomallei* sequence identity increases to 98.2%, indicating that such potential antibody targets are almost entirely conserved across pathogenic *Burkholderia* species (Figure 2a,b; Table 3). A significant caveat to this is that BamA’s conservation is so high that its use as a vaccine antigen may impact the healthy gut microbiome, as has been reported when mice were immunized with *A. baumannii* BamA [166]. Su et al. did address this concern to some extent, showing that the *B. pseudomallei* BamA immunization serum did not cross-react to *E. coli* [27], though future studies should expand on this by directly looking at potential changes in the gut microbiome of vaccinated mice. Despite this potential drawback, revisiting BamA as a Bcc cross-reactive antigen appears justified.

### 7.2. OmpW1 (BPSL1552)

Another outer membrane β-barrel protein that has been explored as a potential *Burkholderia* subunit vaccine antigen is a member of the OmpW family of proteins. Unlike the other proteins discussed in this review, this protein was first identified in Bcc species and subsequently tested as a Bcc-specific vaccine antigen. McClean et al. identified this antigen upon performing a screen for *B. cenocepacia* and *B. multivorans* proteins involved in adhesion to cystic-fibrosis-like lung epithelia cells (CFBE41o-) [173]. They demonstrated that expression of the *B. multivorans* strain LMG13010 OmpW-like protein in *E. coli* resulted in significantly increased adherence to CFBE41o- cells, and that pooled serum from Bcc-colonized cystic fibrosis patients was reactive to the protein. Intraperitoneal immunization of BALB/c mice with this protein and alum, followed by intranasal challenges with either *B. cenocepacia* strain BC7 or *B. multivorans* strain LMG13010, resulted in a significant reduction in lung colonization, but not spleen colonization. The authors also demonstrated that splenocytes from immunized animals could be recalled with recombinant OmpW1, resulting in secretion of IFN-γ, IL-4, IL-17, and IL-6 [173]. This would indicate that Th1, Th2, and Th17 responses were elicited by the protein, a somewhat unexpected finding given the highly Th2-skewing properties of alum.

Given these promising results, as well as the relatively high level of sequence similarity to the *B. pseudomallei* homolog, it was explored whether the *B. pseudomallei* homolog was protective against a *B. pseudomallei* challenge. BALB/c mice were immunized intraperitoneally with the protein homolog and either SAS or alum, and were subsequently challenged intraperitoneally with *B. pseudomallei* strain 576. OmpW1 paired with either alum or SAS did not confer any long-term protection, though the group that received OmpW1 with SAS did experience a delay in median time of death. However, the SAS-alone control suggests this was primarily due to effects of the adjuvant rather than the antigen, as they also had a marked increase in median time of death. A subsequent experiment using an i.p./i.p. model in C57BL/6 mice demonstrated more promising results, with OmpW1 + SAS affording 75% protection on day 80 compared to 12.5% in the adjuvant alone group [174]. Splenocytes from vaccinated animals secreted significant levels of IFN-γ upon ex vivo stimulation with OmpW1, suggesting a strong Th1 response was mounted [175]. A notable downside of these experiments is that the study reported significant issues with protein solubility that could only be overcome by solubilizing the protein in urea. Running their protein on an SDS-PAGE gel without urea resulted in significant smearing, which may be indicative of protein aggregation [174]. Given this information, it is possible that the somewhat lackluster results from protection studies in BALB/c mice was a direct result of the protein not being in its native-like, unaggregated state. Such issues could potentially be addressed by the inclusion of a stabilizing detergent or other membrane-mimicking platform. Overcoming these potential technical problems appears warranted, as OmpW1 is another one of the highly conserved proteins between Bpc and Bcc species. This protein was identified in all the Bcc strains in our dataset, and exhibited an average Bcc-to-*B. pseudomallei* sequence identity of 78.3% and similarity of 88.0% (Table 1). However, when this analysis was restricted to only the extracellular loops, sequence identity drops markedly to 67.7%, potentially limiting this protein’s usefulness as a cross-reactive antibody target (Figure 2c,d; Table 3).

### 7.3. OpcP (BPSS0879), OpcP1 (BPSS0708), and OmpW2 (BPSL2704)

As mentioned already, our group has bioinformatically identified a variety of *B. mallei*-derived proteins that are predicted to be highly antigenic and surface-exposed [69]. We have tested a handful of these proteins as gold nanoparticle-coupled LPS glycoconjugate carrier proteins, including three general β-barrel proteins: OpcP, OpcP1, and OmpW2 [70,71]. Very little is known about these proteins, though OpcP has been functionally characterized as a general diffusion porin [176]. All three of the *B. mallei* strain ATCC 23344-derived β-barrel glycoconjugate constructs elicited robust LPS-specific IgG titers, confirming the proteins’ T cell antigenic properties [70,71]. When these various glycoconjugate vaccine formulations were tested for their ability to protect against a low dose of intranasally delivered *B. mallei* strain ATCC 23344, we found that the OpcP and OmpW2 constructs were the only ones that elicited complete protection, though statistically significant differences were not observed between them and the other constructs. In both groups, more than half the surviving animals did not exhibit lung colonization at the endpoint of the study. The OpcP1 construct only performed marginally worse at 77% protection [70]. When the challenge dose was increased, OpcP and OmpW2 both elicited significant levels of protection with a 77% survival rate. Most of the surviving animals exhibited colonization of the lungs, liver, and spleen of these animals, suggesting dissemination was not prevented [70]. It is worth noting that while *B. mallei* strain ATCC 23344 harbors the *opcP1* gene, 21 of the 30 B. mallei isolates queried in our bioinformatic screen did not, calling into question OpcP1’s potential for use in a glanders vaccine (Table 1). Whether this is related to OpcP1’s relatively reduced performance compared to OpcP and OmpW2 is currently unclear.

We have also used these constructs in a similar immunization scheme followed by a heterologous challenge with *B. pseudomallei* strain K96243. In contrast to the findings in the *B. mallei* challenge model, the OmpW2 construct failed to elicit any meaningful protection against melioidosis [71]. The OpcP and OpcP1 constructs, on the other hand, were capable of conferring protection to a lethal *B. pseudomallei* challenge at 90% and 30% survival, respectively. Furthermore, protection elicited to these constructs was mostly sterilizing, with most animals having no detectable bacteria in the lungs, liver, and spleen at the experimental endpoint. Upon repeating this heterologous challenge study, vaccination with the OpcP or OpcP1 constructs resulted in 90% and 80% protection, respectively. Organ colonization data were like the first trial, though sterilizing immunity was not achieved in the lungs [71]. Because the other constructs tested in this study failed to confer protection to *B. pseudomallei* despite robust LPS-specific antibody titers, it is believed that OpcP- and OpcP1-specific immune responses play a significant role in protection to respiratory melioidosis. Our lab is still investigating whether such protection is mediated by protein-specific antibody or T cell responses; however, we noted that splenocytes recalled with OpcP or OpcP1 elicited significant levels of IFN-γ, IL-17, and IL-2, indicating robust Th1 and Th17 immunity had been elicited by the vaccines [71].

OpcP and OpcP1 are also notable for being moderately well-conserved in Bcc species, both having ~76% Bcc-to-*B. pseudomallei* sequence identity and ~85% similarity (Table 1). OpcP is also reportedly one of the most highly expressed porins in *B. cepacia* and *B. cenocepacia* [177,178], suggesting it is highly expressed for immune recognition. However, the conservation of the extracellular loops of these proteins is considerably lower at 69.8% identity for OpcP1 and 59.8% for OpcP (Table 3). While OmpW2 and its extracellular loops are remarkably well conserved across most Bcc species, *B. multivorans* notably lacks the gene (Table 1 and Table 3). Given that *B. multivorans* is one of the most clinically relevant species in the Bcc, OmpW2’s potential as a universal *Burkholderia* vaccine candidate is likely limited.

### 7.4. Bucl8 (BPSL1972)

Resistance nodulation cell division (RND) efflux pumps are tripartite complexes that play a significant role in extruding clinically relevant antibiotics in Gram-negative bacteria. These pumps assemble as homotrimers and span the outer membrane with a trimeric β-barrel motif. Given their role in antibiotic resistance, they are particularly popular targets for therapeutic development. Three such pumps have been described in *B. pseudomallei* and *B. mallei* [179], though none of these have been tested as vaccine targets. However, one group has recently identified a putative new RND efflux pump, Bucl8, in a screen for *B. pseudomallei* and *B. mallei* proteins containing collagen-like motifs of repeating GXY residues [180]. While it was subsequently determined that these collagen-like motifs allow Bucl8 to bind fibrinogen, it was not determined whether this plays any role during infection [181]. Fusaric acid is currently the only known substrate of Bucl8 [181].

Likely recognizing the inherent difficulty of recombinantly expressing a multimeric integral membrane protein, the study instead opted to try a peptide-based vaccination strategy [182]. They first used ElliPro to bioinformatically predict that the Bucl8 extracellular β-barrel loops could serve as linear B cell epitopes for antibody binding. Then, immunization studies were performed with each of Bucl8’s two extracellular loop peptides chemically conjugated to an inactivated diphtheria toxoid carrier protein and formulated in AddaVax. These vaccines, delivered subcutaneously, were capable of eliciting robust IgG1-skewed antibody titers that reacted to the respective peptide or an attenuated *B. pseudomallei* strain [182]. However, this subcutaneous immunization was not protective against a heterologous intranasal challenge of 8 × 10^6^ CFU of *B. thailandensis* strain E264 [183]. This result can likely be attributed at least in part to the lack of mucosal immunity that would be expected from such an immunization scheme. A follow-up experiment was performed in which CD-1 mice were intranasally immunized with diphtheria toxin linked to one of the two Bucl8 peptides and fluorinated cyclic diguanosine monophosphate, a STING agonist that has not been previously used as an adjuvant in any *Burkholderia* vaccine studies. These animals were challenged intranasally with 8 × 10^5^ CFU of *B. thailandensis* strain E264 expressing a luciferase gene. The vaccinated animals had significantly reduced weight changes, significantly lower luminescent signals at the site of infection at day seven, and significantly lower bacterial burdens in the lungs and nasal wash at days three and seven, respectively [183]. While significant differences in protection were not reported, this is not entirely surprising given that the studies were done in an outbred mouse model using a heterologous challenge and only immunizing with a single peptide epitope. The fact that any protection at all was observed in such a model is quite promising, and it would be interesting to see how this antigen performs when administered as a full protein or against a *B. pseudomallei* challenge. Future studies should confirm that Bucl8 vaccination is protective in more clinically relevant models of glanders, melioidosis, or even Bcc colonization of the lung. Bcc cross-protection studies may be particularly warranted given that Bucl8 is broadly conserved in Bcc species and exhibits a relatively high level of sequence conservation in the extracellular loops at 88.0% sequence identity (Table 1 and Table 3).

**Table 3 vaccines-12-00313-t003:** Outer Membrane β-barrel antigen sequence identity in extracellular loop domains.

*B. pseudomallei* K96243 Locus	Protein Name	Bcc % Identity to K96243 (Full Protein)	Bcc % Identity to K96243 (Extracellular Loops)
BPSL2151	BamA	94.0	98.2
BPSL2704	OmpW2	87.8	88.9
BPSL1972	Bucl8	78.0	88.0
BPSS0708	OpcP1	75.7	69.8
BPSL1552	OmpW1	78.3	67.7
BPSS0879	OpcP	75.4	59.8

## 8. General Outer Membrane Proteins: Lipoproteins

Besides the integral β-barrel proteins, lipoproteins make up the other major class of proteins expressed on the Gram-negative outer membrane. Because most lipoproteins are expressed on the cytoplasmic membrane or on the inner leaflet of the outer membrane, lipoproteins are a relatively underappreciated class of potential antibody targets in Gram-negative bacteria. While early studies identified a handful of surface-exposed lipoproteins in *E. coli* [184,185], *Treponema pallidum* [186], and *Borrelia burgdorferi* [187], such examples were generally regarded as anomalous. However, recent evidence suggests that such surface-exposed lipoproteins are much more widespread than previously believed, and thus represent an emerging field of study that is particularly relevant for vaccine researchers working with Gram-negative bacteria [188]. While outer membrane lipoproteins are relatively easy to identify bioinformatically due to their differences in signal sequence, it is not so easy to determine which outer membrane lipoproteins are surface exposed, and instead they must be identified empirically. Schell et al. performed studies to characterize the outer membrane proteome of *B. pseudomallei* and *B. mallei*, and identified several lipoproteins in the process, but the approach employed would not necessarily distinguish between surface- and periplasm-facing lipoproteins [189]. However, Sousa et al. recently performed surface trypsin shaving studies of *B. cenocepacia* strain J2315 and identified 55 surface-exposed proteins [190]. When we analyzed these proteins further using DTU Health Tech’s SignalP v6.0 program [191], we found that 11 were predicted lipoproteins. These proteins corresponded to the following locus tags: BCAL0340, BCAL1493, BCAL2166, BCAL1893, BCAL0151, BCAL2687, BCAL3204, BCAL2645, BCAL1288, BCAL1881, and BCAL0304. The BCAL3204 predicted lipoprotein is particularly noteworthy, as the *B. pseudomallei* homolog is extremely well-conserved and has shown great promise as a melioidosis vaccine antigen [192,193,194]. Another of these proteins, BCAL2645, has yet to be used in a *Burkholderia* vaccine to date, though it seems a particularly strong candidate given some of its properties which will be discussed here.

### 8.1. Omp7 (BPSL2765)

The first study of a *Burkholderia* lipoprotein antigen was performed by Hara et al. [192]. The authors bioinformatically identified 12 OmpA family proteins (designated Omp1-12) and attempted to purify them as recombinant antigens to test for melioidosis serum reactivity and as subunit vaccine antigens. Of these 12 proteins, 5 are predicted by SignalP v6.0 to be lipoproteins including Omp1 (BPSL0999), Omp4 (BPSL2062), Omp5 (BPSL1659), Omp7 (BPSL2765), and Omp9 (BPSS0909) [191]. Only 6 of the 12 proteins were successfully cloned from *B. pseudomallei* strain D286 and expressed in *E. coli*, including the putative lipoproteins Omp4, Omp5, and Omp7. Omp7, as well as the non-lipoprotein Omp3 (discussed later), reacted particularly strongly to human melioidosis serum, and, as a result, were down-selected for testing as vaccine antigens. BALB/c mice were immunized intraperitoneally with Omp3 or Omp7 in Fruend’s adjuvant before challenging i.p. with 10 × LD_50_ of *B. pseudomallei* strain D286, resulting in 50% protection in both vaccination groups compared to 0% in the control animals [192].

Given these promising results, Omp7 has been utilized in subsequent vaccination studies. One such study used a similar model as Hara et al., albeit with a much higher challenge dose of 100 × LD_50_ and using SAS as the adjuvant [193]. In this study, Omp7 was pooled with a trio of T cell antigens associated with chronic infection: IscA, TadE, and AcoD (discussed later). This antigen pool was further mixed with either the well-studied T cell antigen LolC (discussed later), the potent B cell antigen CPS, or neither. Between 30 and 50% protection level was observed, depending on the exact combination of antigens, compared to 0% in the control animals [193]. However, given the complex nature of these formulations, it is impossible to ascertain to what extent Omp7 was responsible for this protection.

More recently, Dyke et al. extensively characterized Omp7, identifying it as a peptidoglycan-associated lipoprotein (Pal) [194]. The authors went on to show that *B. mallei* Omp7 is required for virulence, with the isogenic mutant being largely attenuated in BALB/c mice. The mutant also appeared to be more susceptible to osmotic stress, polymyxin B, and killing by macrophages, potentially due to disruptions in the cell envelope, given Pal’s known role in cell envelope stability in other species [194]. The study also demonstrated that intranasal vaccination with *B. mallei*-derived Omp7 expressed by a PIV5 vector elicited 80% long-term protection to an intranasal challenge with 10 × LD_50_ *B. mallei* strain ATCC 23344, though sterilizing immunity was not achieved in most animals, with both lungs and spleens harboring bacteria [194]. Studies outlining the mechanism of protection elicited by Omp7 are somewhat sparse, though a study by Gourlay et al. provides some insight [30]. Generating antibodies against a specific epitope of Omp7, they demonstrated that such antibodies promoted the killing of *B. pseudomallei* by neutrophils and caused the bacteria to agglutinate [30]. While Pal-like proteins are generally thought to be found on the periplasmic side of the outer membrane, these functional antibody studies may indicate that some proportion of Omp7 is located on the extracellular face. This would align with studies of Pal from *E. coli* [195] and *Haemophilus influenzae* [196], which have been shown to be expressed on both the periplasmic and extracellular side of the outer membrane.

Running in parallel with the Bpc studies, the *B. cenocepacia* homolog BCAL3204 has also been intensely studied. This protein is one of the most highly immunodominant B cell antigens in Bcc species, being readily recognized by sera from human CF patients with Bcc colonization of the lung [197]. This was confirmed by Makidon et al. by performing immunization studies with a nanoemulsion containing outer membrane proteins purified directly from *B. cenocepacia*. One of the most immunodominant proteins in this formulation was determined to be BCAL3204 [198]. Subsequent studies characterizing the protein found that an isogenic *B. cenocepacia* mutant had increased susceptibility to polymyxin B, caused reduced IL-8 secretion in CFBE41o- cells, and had a reduced ability to adhere to CFBE41o- cells [199]. Additionally, it has been demonstrated that sera from human CF patients with Bcc colonization of the lung is cross-reactive to peptides derived from *B. pseudomallei* Omp7, providing strong empirical evidence for the use of this antigen as a pan-*Burkholderia* vaccine antigen [200]. Our own screen for potential cross-reactive antigens adds additional support, as Omp7 is the third most conserved *Burkholderia* surface antigen discussed in this review, with the Bcc homologs exhibiting an average of 85.5% identity and 93.1% similarity to the *B. pseudomallei* strain K96243 ortholog (Table 1; Figure 3a,b).

### 8.2. Omp1 (BPSL0999)

While Hara et al. was ultimately unable to express Omp1 [192], nor has it ever been used in a published *Burkholderia* vaccine study to date, recent evidence in support of it warrants discussion. This protein was first identified as a seroreactive marker in high-throughput screens of sera from human melioidosis patients [75,76] and a human glanders patient [201]. Additionally, Omp1-derived peptides were found to be some of the most robust stimulators of IFN-γ production by PBMCs derived from melioidosis seropositive human donors, with nearly all donors responding to the stimulus [202]. The *B. cenocepacia* homolog, BCAL2645, is another one of the lipoproteins identified via Sousa et al.’s surface trypsin shaving study, providing evidence that it is a potential surface-exposed target for antibody binding [190]. Further, Seixas et al. recently demonstrated that polyclonal antibodies towards BCAL2645 impair adhesion and invasion of CFBE41o- cells by *B. cenocepacia* [203]. The ability of these antibodies to impede adherence suggests a possible mechanism of protection if this antigen were to be used in vaccination. Additionally, Omp1 is the second most well-conserved surface antigen between Bpc and Bcc species discussed in this review, following only BamA (Table 1). The Bcc homologs of this protein appear to have an average sequence identity of 93.4% and similarity of 95.9% when compared to the *B. pseudomallei* strain K96243 sequence (Table 1; Figure 3c,d). Unlike BamA, Omp1 does not appear to be widely conserved across Gram-negative species, making it far less likely to have an impact on the healthy human microbiome. Together, this implicates Omp1 as a top candidate for a pan-*Burkholderia* vaccine, assuming protective efficacy can be demonstrated in an active immunization model.

## 9. ABC-Binding Cassette (ABC) Transporters

ABC transporters are a superfamily of multicomponent protein complexes found in all kingdoms of life whose purpose is to transfer various cargo molecules across cell membranes. ABC transporters share a similar architecture consisting of two hydrophobic integral membrane domains and two highly conserved hydrophilic nucleotide-binding domains within the cytoplasm that hydrolyze ATP to provide energy to the system [204,205,206]. In Gram-negative bacteria, ABC importers also associate with a periplasmic receptor protein that binds the cargo prior to transport, while ABC exporters associate with a membrane fusion protein and an outer membrane factor that serve as a pore for cargo to reach the extracellular environment [204,205,206]. In bacteria, ABC systems are known to play vital roles in survival and virulence with functions relating to nutrient uptake, toxin/antibiotic efflux, lipoprotein localization to the outer membrane, and siderophore secretion [206]. From an immunological perspective, only the outer membrane factor component of ABC exporters is exposed to the extracellular space for antibody binding. All other components of ABC transporters in Gram-negative bacteria are localized to the periplasm, inner membrane, or cytoplasm, and would therefore be expected to act primarily as T cell antigens when used as vaccine antigens. To our knowledge, there are currently no examples of vaccination with the outer membrane factor of a *Burkholderia* ABC exporter.

Harland et al. created a comprehensive annotation of *B. pseudomallei* and *B. mallei* ABC proteins [207]. Based on this annotation, they selected three ABC transporter proteins to test as potential subunit vaccine antigens: LolC (BPSL2277), PotF (BPSS0467), and OppA (BPSS2141) [208]. LolC is one of the integral membrane components of the LolCDE ABC exporter which has been extensively studied in *E. coli* [209]. This ABC exporter is essential to most Gram-negative species, as it is required for transporting lipoproteins from the inner membrane to the outer membrane [209]. PotF is the periplasmic receptor protein of PotFGHI, an ABC importer of putrescine [210]. Notably, *B. mallei* has lost the direct homolog of this *potF* gene through genome reduction. Lastly, OppA is the periplasmic receptor protein of the OppABCDF importer of oligopeptides. Such oligopeptides serve as an amino acid and carbon source for multiple species of bacteria [211,212]. To demonstrate the proteins’ potential as T cell antigens, the group challenged naïve BALB/c mice with *B. pseudomallei* strain 2D2 and stimulated splenocytes recovered from the animals’ ex vivo with recombinant LolC, PotF, or OppA. Antigen-recalled splenocytes secreted detectable levels of IFN-γ, suggesting robust Th1 responses to these proteins over the course of natural infection, a finding that has been replicated by others [208,213,214,215].

Harland et al. also demonstrated the antigens’ protective potential by intraperitoneally vaccinating BALB/c mice with each individual antigen paired with monophosphoryl lipid A-trehalose dicorynomycolate as an adjuvant before challenge via the i.p. route with 40 × LD_50_ of *B. pseudomallei* strain K96243. Mice immunized with PotF or LolC displayed 50% and ~85% survival, respectively, while OppA vaccination afforded only ~20% protection [208]. In a follow-up experiment in which LolC was paired with different adjuvants, the authors reported that the same LolC/adjuvant combo resulted in only 33% protection, though this could be attributed to the higher challenge dose of 70 × LD_50_. The other adjuvants tested with LolC included various combinations of Emulsigen, CpG ODN 10103, and AbISCO-100, all of which were administered subcutaneously instead of intraperitoneally. The protection afforded by these different formulations ranged from 17 to 66% with the higher challenge dose [208].

A subsequent study performed by Scott et al. found that LolC paired with Alhydrogel and CpG ODN 2006 did not protect BALBc mice from mortality when administered subcutaneously followed by an intraperitoneal challenge [216]. However, they did report a significant increase in median time of death from 3.5 to 10 days, suggesting at least some level of protection afforded by the antigen. This group also found that LolC paired with *B. pseudomallei* CPS provided slightly improved protection and reduced disease severity compared to mice vaccinated with CPS alone, though statistically significant differences were not reported. That said, vaccination with CPS alone was significantly more protective than vaccination with LolC alone, again supporting the dominant role of humoral immunity in protection in murine models of respiratory infection [216].

Finally, Whitlock et al. compared LolC’s protective efficacy to two other *Burkholderia* subunit antigens, BimA and BopA, in a clinically relevant intranasal immunization and challenge model. Using a cationic liposome-DNA complex as an adjuvant, the results mirrored that of Scott et al., finding that none of the LolC-immunized mice survived a 2 × LD_50_ challenge with *B. pseudomallei* strain 1026b. They likewise found that LolC-vaccinated mice had an increase in median time of death, with all mice surviving for 1–7 weeks compared to the naïve control group which all succumbed within a week of infection. By comparison, mice immunized with BimA and BopA were approximately 20% and 60% protected with similarly increased median times of death [82]. Interestingly, when the authors attempted a heterologous challenge with *B. mallei* strain ATCC 23344, LolC-vaccinated mice were 83% protected [82]. These data would suggest that LolC is only mildly protective in melioidosis models, though perhaps shows promise as a protective glanders antigen. Because of its essential function in shuttling lipoproteins to the outer membrane, LolC is unsurprisingly one of the most highly conserved antigens between Bpc and Bcc species discussed in this review at 93.3% sequence identity and 97.1% sequence similarity (Table 2). As such, LolC may be another candidate for the development of a pan-*Burkholderia* vaccine.

## 10. Miscellaneous Protein Antigens

### 10.1. Omp3 (BPSL2522)

In the Hara et al. study mentioned previously, they identified two OmpA domain-containing proteins, Omp3 and Omp7, that reacted strongly to human melioidosis serum and conferred similar levels of protection in i.p./i.p. vaccination/challenge murine models [192]. Despite this, only Omp7 received further attention as a potential candidate vaccine antigen while Omp3 has only seen use as a serodiagnostic marker, with numerous studies identifying it as highly reactive to human melioidosis serum [75,76,77,192,217,218] and horse glanders serum [140]. In addition to these B cell antigenic properties, PBMCs derived from healthy melioidosis seropositive donors and recovered melioidosis donors react strongly to Omp3-derived peptides, suggesting that robust T cell immunity is also mounted to this protein during natural exposure to *B. pseudomallei* [202].

Despite lacking typical motifs observed in outer membrane proteins of Gram-negative bacteria (i.e., β-barrel domain or lipobox sequence), there is evidence to suggest that this protein is expressed on the outer membrane surface and is therefore a potential target for humoral immunity. Schell et al. identified Omp3 in trypsin shaving studies of *B. pseudomallei* and *B. mallei* outer membrane preparations, confirming it as an outer-membrane-associated protein [189]. Additionally, the highly conserved *B. cenocepacia* homolog, BCAL2958, is one of the 55 proteins identified in Sousa et al.’s surface trypsin shaving studies, strongly suggesting it is extracellularly oriented, at least in *B. cenocepacia* [190]. Looking closer at the primary sequence of this protein, we noted the presence of a patch of hydrophobic tryptophan residues near the N-terminus. Such tryptophan-rich patches are known to occasionally serve as membrane anchoring motifs [219], providing a potential explanation for the proteins apparent surface localization.

The *B. cenocepacia* Omp3 homolog has gained attention in recent years as a potential therapeutic target for Bcc colonization of the CF lung. Two separate studies investigating this protein demonstrated that it is highly reactive to serum from CF patients that are positive for Bcc lung colonization [190,220]. They also noted that antibodies to *B. cenocepacia*-derived Omp3 are highly cross-reactive to 12 strains of Bcc from 7 distinct species, suggesting it is highly conserved [220]. In fact, our own analysis confirms that Omp3 is extremely well-conserved not just among all the Bcc species in our dataset, but also between the Bpc and Bcc groups (Table 1). Of all the putative targets of humoral immunity discussed here, Omp3 ranks as having the fourth highest degree of sequence conservation, with the Bcc homologs of Omp3 exhibiting an average of 91.6% sequence identity and 92.9% sequence similarity to Bcc homologs (Table 1; Figure 3e,f). It has also been demonstrated that recombinant *B. cenocepacia*-derived Omp3 is highly stimulatory to healthy human primary neutrophils, causing them to release significantly higher levels of TNFα, elastase, nitric oxide, catalase, and myeloperoxidase [220]. The mechanism and purpose of this function is still unknown. Considering how highly conserved this protein is across *Burkholderia* species, it seems likely that it plays some other role related to survival in the environment or cell envelope integrity/barrier function. The immunodominance of this protein in Bpc- and Bcc-associated diseases, its surface localization, and its high degree of Bpc-to-Bcc sequence conservation make it a key target for a pan-*Burkholderia* vaccine.

### 10.2. MprA (BPSS1993)

MprA is a secreted serine protease generally regarded as the most active secreted protease produced by *B. pseudomallei* [221,222,223]. However, *mprA* expression appears to be primarily restricted to the stationary growth phase, and the encoded enzyme does not appear to play any significant role in virulence in mouse models of infection [222]. Furthermore, melioidosis patient sera is only mildly reactive to the protein, with less than half of samples reacting to it in one study [204]. As such, MprA may not be an ideal target for neutralizing antibodies. Chin et al. immunized BALB/c mice with MprA in Freund’s adjuvant, resulting in high antigen-specific antibody titers [223]. Animals challenged i.p. with 10 × LD_50_ *B. pseudomallei* strain D286 had significantly increased survival times compared to the control animals, but succumbed to infection past day 25. These animals exhibited bacteremia, splenomegaly, and abscess formation after the challenge, indicating that the bacteria were not prevented from disseminating [223]. As such, MprA appears to confer some low level of protection, at least in i.p. models of infection in mice, but is overall less protective than many of the other antigens discussed in this review. Furthermore, an MprA homolog was not identified in any of the *B. mallei* proteomes in our dataset, nor was it identified in any Bcc species except *B. stagnalis* and *B. ubonensis*, neither of which are particularly relevant to human disease. As such, MprA has little potential as a glanders or Bcc cross-reactive antigen.

### 10.3. AhpC (BPSL2096)

Alkyl hydroperoxide reductases such as AhpC in *B. pseudomallei* are frequently upregulated in bacteria during the intracellular phase of infection to survive host-derived hydrogen peroxide [224]. As a result, natural exposure to such intracellular bacteria is often associated with robust immune responses to these proteins, making them popular candidate antigens for a variety of pathogenic bacteria [225,226,227]. This trend is also observed with *B. pseudomallei* AhpC, which acts as a highly serodiagnostic antigen in human melioidosis [75,141,228] and horse glanders [140]. PBMCs from human melioidosis seropositive donors also respond strongly to AhpC-derived peptides or full-length protein and the strength of this response appears to correlate with improved disease outcomes [141,202,228,229]. Together, these studies suggest that AhpC is strongly immunogenic during natural melioidosis infection, eliciting robust B and T cell responses. Furthermore, AhpC is extremely well-conserved and has the highest level of sequence identity between the Bpc and Bcc of any antigen discussed here (Table 2), making it a very promising T cell antigen for a Bcc cross-reactive vaccine.

Schmidt et al. performed a vaccination study using AhpC paired with the CPS-CRM197 glycoconjugate antigen [32]. To account for AhpC’s enzymatic activity and any detrimental effect it might have on immunity or animal health, the authors introduced a cysteine-to-glycine substitution at residue 57 which renders the enzyme nonfunctional. C57BL/6 mice were immunized subcutaneously with the AhpC mutant and CPS-CRM197, along with CpG ODN 2006 and Alhydrogel as adjuvants. This resulted in robust IgG1 and IgG2a/b titers, suggesting a Th1/Th2-balanced response [32]. These antibodies were functionally characterized via opsonophagocytosis assay and were capable of significantly increasing the uptake of *B. thailandensis* by macrophages, though this was largely attributed to CPS-specific antibodies [32]. Splenocytes from immunized animals were recalled with full-length AhpC^C57G^ or AhpC-derived peptides, resulting in significant induction of IFN-γ, IL-5, and IL-17 [32]. Finally, in mouse challenge studies, the AhpC^C57G^ and CPS-CRM197-vaccinated animals exhibited 70% protection against an intranasal challenge of *B. pseudomallei* strain K96243 compared to 0% in the adjuvant group. A follow-up study further confirmed that AhpC elicits particularly robust T cell responses after vaccination, though they were unable to establish if this had any role in protection [142]. On the contrary, mice immunized with Hcp1 + AhpC + CPS-CRM197 were less protected than those immunized with Hcp1 + CPS-CRM197, albeit not to a statistically significant degree [142]. Finally, it is also worth noting that AhpC vaccination has reportedly been associated with reactogenicity after exposure to *B. pseudomallei* [230], which could significantly limit the usefulness of the antigen due to safety concerns.

### 10.4. IscA (BPSL2287), TadE (BPSL1897), and AcoD (BPSL3369)

Because many *B. pseudomallei* vaccines fail to confer sterilizing immunity in mice resulting in chronic infection, Champion et al. used qRT-PCR to identify upregulated genes in chronically infected mouse tissue [193]. Six such genes were identified, but only IscA, TadE, and AcoD could be recombinantly expressed. These genes are notably predicted by pSORTb to be cytoplasmic or in the inner membrane [193], meaning they are likely functioning primarily as T cell antigens in a subunit vaccine. These antigens were mixed with Omp7 and administered intraperitoneally to C57BL/6 mice as a cocktail paired with LolC, CPS, or alone. These antigens were adjuvanted with SAS and the vaccine was administered i.p. before challenging i.p. with 100 × LD_50_ of *B. pseudomallei* strain K96243. All formulations containing the chronic antigen cocktail, regardless of CPS or LolC presence, exhibited 30–50% long-term protection compared to 0% in naïve animals [193]. However, because Omp7 was included in all reported formulations, it cannot be said whether the reported protection was due to the inclusion of the chronic disease-associated T cell antigens, the putative B cell antigen Omp7, or a combination of both. Another significant limitation of this study is that organ colonization was not reported for the surviving animals, so it is unclear if these antigens were efficacious in preventing chronic infection in the surviving animals, which was ultimately the goal of this study. If such concerns could be addressed, these proteins could serve as potentially promising T cell antigens in a Bcc cross-protective vaccine, given their high degree of sequence conservation in Bcc species, particularly IscA (Table 2).

## 11. Polysaccharides

Of all the subunit antigens assessed in Bpc vaccines, the capsule polysaccharide (CPS) and lipopolysaccharide (LPS) are by-and-large the most well-studied. Those interested in an in-depth analysis of these antigens are recommended to consult this recent review [231]. Polysaccharide antigens are considered T-cell-independent antigens because they cannot be efficiently loaded onto classical MHC molecules for subsequent activation of T cells. On their own, such polysaccharide antigens are essentially incapable of inducing T-cell-dependent B cell activation, resulting in low affinity antibodies and minimal antibody class-switching. Glycoconjugate vaccines circumvent this issue by covalently attaching the polysaccharide to a carrier protein that is capable of being loaded onto MHC. Resultant T cell responses are then able to activate germinal centers harboring polysaccharide-specific B cells, resulting in higher affinity class-switched antibodies [232].

*Burkholderia* polysaccharide vaccines include some that have forgone a carrier protein altogether [68,69,193,233,234,235], while others have opted to include non-*Burkholderia*-derived [31,68,69,142,216,235,236,237,238,239,240] or *Burkholderia*-derived [64,67,68,69,70,71] carrier proteins. Furthermore, the polysaccharides themselves come from various sources with different groups using *B. pseudomallei*-derived polysaccharides [31,64,142,216,233,236,237], non-pathogenic *B. thailandensis*-derived polysaccharides [67,68,69,70,71,193,234], recombinantly expressed polysaccharides [238,241], or synthetic polysaccharides [235,239]. LPS-specific but not CPS-specific antibody titers are a known correlate of protection in human melioidosis [73]. Despite this, both LPS- and CPS-specific antibodies have been shown to be protective in rodent models, with in vitro studies indicating that such antibodies promote bacterial opsonophagocytic killing in a complement-dependent manner [26,28,29,31,68,69,216,242,243].

While vaccine studies have almost exclusively focused on one variant of LPS (LPS-A) and one variant of CPS (CPS-I), *B. pseudomallei* and *B. mallei* express other distinct variants of these polysaccharides and harbor strain- and species-specific expression patterns. As much as 10% of *B. pseudomallei* isolates solely express LPS-B/B2, which harbors an entirely different O-antigen backbone structure and is non-reactive to LPS-A serum [244,245,246]. *B. mallei* is only known to express LPS-A, though its O-antigen differs in acetylation patterns, leading to subtle differences in antibody reactivity [239,242,247,248]. Similarly, while vaccine studies have primarily focused on CPS-I, there are three other putative capsular polysaccharide synthesis loci, and at least one distinct form of CPS has been described (CPS-II) [249,250]. To date, the relative importance of LPS-B/B2 and CPS-II as vaccine antigens remains largely unexplored, and may hint that these polysaccharide antigens have limitations in their ability to broadly protect against all *B. mallei* and *B. pseudomallei* isolates, let alone Bcc species. In fact, there is little evidence to suggest that such polysaccharide antigens would have much applicability as Bcc cross-reactive antigens, with only some isolates of *B. cepacia, B. ubonensis*, and *B. stabilis* exhibiting polysaccharide-specific serum cross-reactivity to Bpc polysaccharides [246,251,252].

## 12. Conclusions and Future Directions

Many of the protective Bpc antigens described herein have only been tested in what are considered outdated vaccination/challenge models, chiefly those that utilized the intraperitoneal or intravenous administration routes. BamA, Omp3, and OmpW1 are three such antigens that have yet to be tested in respiratory melioidosis and glanders models, despite showing significant protective efficacy in i.p. models [27,173,174,192]. Additionally, these antigens exhibit a significant degree of amino acid sequence conservation across both the Bpc and Bcc, making them key candidates for a pan-*Burkholderia* vaccine. Future studies should explore whether their protective capabilities extend to respiratory melioidosis and glanders infection models, which are of greater significance from a public health standpoint. If such studies show promise, then research into whether BamA, Omp3, or OmpW1 can elicit cross-protection to Bcc species would be warranted.

We also noted numerous reports of solubility, aggregation, and expression issues for certain antigens, particularly transmembrane β-barrels like BamA, OpcP, and OmpW1 [27,70,71,173,174]. Aggregated or misfolded proteins are significantly impaired in their ability to elicit antibodies to conformational epitopes, thus lowering the likelihood of observing protection when immunizing with the purified protein. Unfortunately, there do not appear to be widely adopted solutions to this problem, as membrane protein stabilizing agents (i.e., detergents, lipid nanodiscs) are not commonly used in vivo, and their ability to maintain a protein’s structure in vivo is unclear. Regardless, if these antigens are to proceed to human trials in any *Burkholderia* vaccine formulation, then such issues will likely need to be addressed.

We were also surprised to find that antigens predicted to serve primarily as T cell antigens generally exhibited only mild or negligible improvements to survival in respiratory challenge models, despite the generally held belief that T cell immunity is critical to controlling the intracellular phase of infection and preventing the bacteria from establishing itself as a chronic disease. While i.p. models have generally shown more promise with such antigens [136,208,216], respiratory challenge models utilizing Hcp1, LolC, and AhpC have only been able to modestly improve survival at best, and have not been clearly demonstrated to prevent long-term organ colonization in mice [31,68,69,70,71,216]. On the contrary, the B-cell-restricted polysaccharide antigens LPS and CPS consistently appeared to be more important for mediating protection in such murine models of respiratory-acquired melioidosis and glanders. Furthermore, adding a T-cell-restricted antigen to a formulation containing CPS or LPS has never been shown to confer a statistically significant improvement to survival or organ colonization [31,68,69,216], though one study came close to demonstrating significantly improved survival [31]. It is unclear whether this represents a limitation of such murine respiratory models in recapitulating the human disease, a failure of these vaccines to elicit sufficient T responses, or whether T cell responses are not as important for vaccine-mediated protection as previously believed. The latter explanation appears unlikely, given the strong correlation between Th1-mediated responses and improved disease outcomes in humans [33,34,141]. Due to the clinical relevance of chronic melioidosis, answering this question would significantly benefit the field.

Overall, our screen for highly conserved *Burkholderia* antigens yielded a few particularly promising candidates for eliciting cross-reactivity to Bcc species including BamA, Omp1, Omp3, Omp7, OmpW1, and OpcP, which are all putative targets of humoral immunity, as well as AhpC and LolC, which are likely T-cell-restricted antigens. While further exploration of these antigens as pan-*Burkholderia* candidate antigens appears justified, we should mention that there are other hurdles to Bcc vaccine development that also need to be addressed. First, there are still currently no widely adopted animal models of Bcc lung colonization that simultaneously recapitulate a CF-like disease state [32]. In the absence of such models, correlates of protection cannot be readily established, making the informed design of a Bcc vaccine particularly challenging. In fact, this has been a long-standing challenge to designing vaccines for CF patients, as the unique environment of the CF lung may impair certain types of immunity from clearing the infection [253]. In general, past attempts to design vaccines specifically for CF patients have shown mostly disappointing results in humans [253], though some *Pseudomonas aeruginosa* vaccines have exhibited a limited ability to protect CF patients, indicating that vaccination may still be a viable approach at prevention [253,254,255]. One recent review points out that many of the prior failed human clinical trials of *P. aeruginosa* vaccines have used platforms and administration routes that would preferentially induce systemic immune responses, not mucosal immunity [253]. The design of mucosal-stimulating vaccines is a highly active and constantly evolving area of research, so it stands to reason that researchers developing vaccines to such CF-associated bacteria will have new avenues to explore as novel strategies emerge for the targeted induction of mucosal immunity.

Despite the remaining challenges, we hope that this review has provided some useful insight into the rational design of subunit and glycoconjugate *Burkholderia* vaccines. Designing a vaccine capable of conferring protection to both Bpc and Bcc species has been alluded to in the past, but the results of our bioinformatic screen appear to lend additional support to this strategy, and hopefully sparks the interest for other researchers to explore pan-*Burkholderia* vaccine research.

## Figures and Tables

**Figure 1 vaccines-12-00313-f001:**
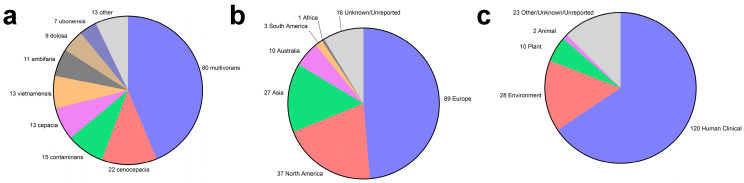
Breakdown of the Bcc isolates used in the BLAST screen for Bcc-to-*B. pseudomallei* antigen sequence conservation based on (**a**) species, (**b**) continent of origin, and (**c**) isolation source. Graphs were created using Graphpad Prism 10.

**Figure 2 vaccines-12-00313-f002:**
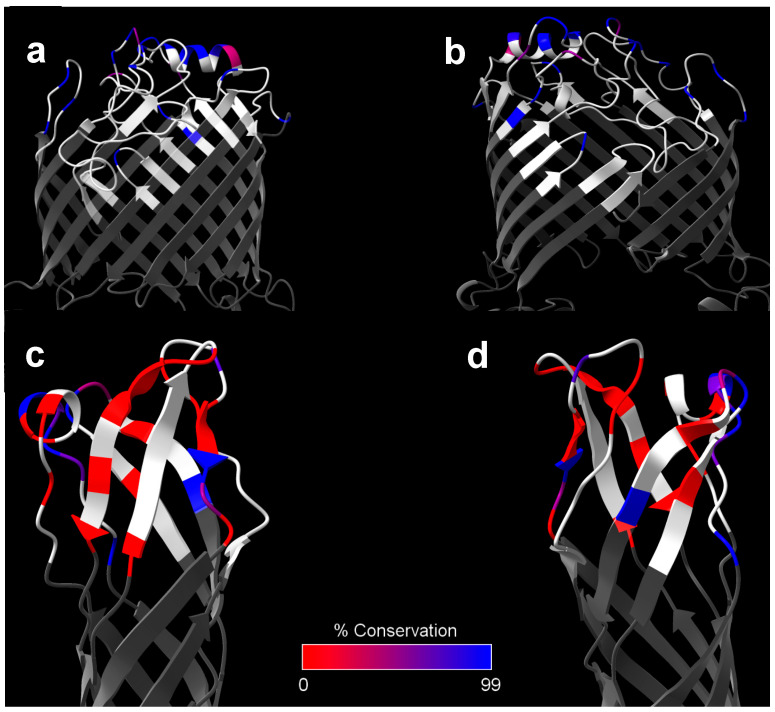
ChimeraX v1.6.1-generated ribbon structures of β-barrel antigens with a high degree of Bcc-to-*B. pseudomallei* amino acid sequence identity [170]. Residues not predicted using DeepTMHMM v1.0.24 to be extracellularly oriented are colored in translucent gray [171]. Extracellularly oriented residues that are 100% conserved between the Bcc homologs and the *B. pseudomallei* K96243 reference are colored light gray. Residues which are substituted or missing in Bcc homologs are colored according to the scale bar to represent the degree of sequence conservation (identity). (**a**,**b**) Front and back views of the BamA extracellular loops (AFDB accession AF-Q63T20-F1) [172]. (**c**,**d**) Front and back views of the OmpW1 extracellular loops (AFDB accession AF-Q63UP4-F1) [172].

**Figure 3 vaccines-12-00313-f003:**
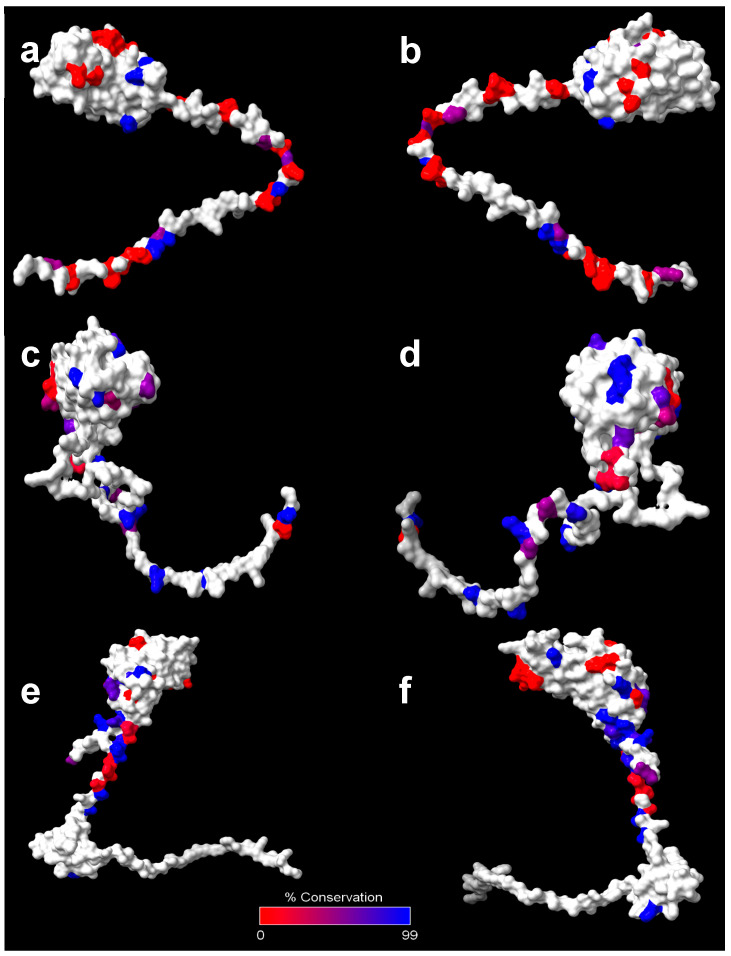
ChimeraX v1.6.1-generated surface structures of OmpA domain-containing antigens [170]. Residues that are 100% conserved between the Bcc homologs and the *B. pseudomallei* K96243 reference are colored light gray. Residues which are substituted or missing in Bcc homologs are colored according to the scale bar to represent the degree of sequence conservation (identity). (**a**,**b**) Front and back views of Omp7 (AFDB accession AF-Q63RA7-F1) [172]. (**c**,**d**) Front and back views of Omp1 (AFDB accession AF-Q63W89-F1) [172]. (**e**,**f**) Front and back views of Omp3 (AFDB accession AF-Q63RZ9-F1) [172]. The OmpA domain, which is the most likely region to be targetable by antibodies, is the globular domain in the upper half of each panel.

## Data Availability

All data are available from published manuscripts. Additional analysis of the data is presented in this review and is available upon request.

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
