# Peer review of "Burkholderia pseudomallei Complex Subunit and Glycoconjugate Vaccines and Their Potential to Elicit Cross-Protection to Burkholderia cepacia Complex"

_vaccines, 2024, doi:10.3390/vaccines12030313_

Round 1

Reviewer 1 Report

Comments and Suggestions for Authors

Overall this is a very well-written and cited review. The writing is clear and it could be published in its current form, however, I think it may be of greater importance if some of the following were addressed.

I think it could benefit from a description of the pathogenesis and diseases caused by each of these species (or groups of species Bpc/Bcc) what is similar and what is different including a figure to orient the readers before going into an in-depth discussion of virulence factors and the development of many as vaccine candidates. For example, are the virulence strategies of Bpc and Bcc similar? Even a figure with each bacteria demonstrating where the location of cross-reactive antigens are would be useful. 

As the entire review deals with glycoconjugate and subunit vaccines, a brief description of how these elicit immune responses would be beneficial. Also, a table of which adjuvants seem to work best in subunit formulations may be beneficial to the readers. 

If the goal was to make a cross-protective glycoconjugate vaccine would you have to include Bcc polysaccharides? 

I think the title is somewhat misleading because a lot of the text is about vaccine candidates that have no potential to elicit cross-protection. With this in mind, this review could be drastically shortened. It may be more pertinent to describe the best vaccine candidates for Bcp and what types of immune responses produce protection. Then specifically address proteins with potential cross-protection and their ability to provide similar protective responses.

The conclusions discuss outdated models of infection, and the introduction could benefit from a description of which models the authors believe are the most appropriate for studying vaccination.

Again the discussion talks about the extracellular phase of infection, I think a description of the basic virulence of each group would be helpful in the introduction and describe if the goal of a vaccine is sterilizing immunity or if the goal is different.

Specific comments:

Lines 627-628 indicate Hcp1 is most likely a T cell antigen. However, Lines 630-632 demonstrate that human sera respond to Hcp1. Can the authors provide a comment on this?

Author Response

Response to reviewers of manuscript ID: vaccines-2850735: “Burkholderia pseudomallei Complex Subunit and Glycoconjugate Vaccines and Their Potential to Elicit Cross-Protection to Burkholderia cepacia Complex”.

We greatly appreciate your editorial suggestions to ensure that our manuscript is in accordance with Vaccines and MDPI guidelines. Additionally, we appreciated the reviewers’ suggestions and criticism to our manuscript. We have enclosed a revised manuscript for your consideration that has been modified in line with all the reviewers’ comments. Find appended below the details of the changes made in response to these comments.

Reviewer 1.

1.)   I think it could benefit from a description of the pathogenesis and diseases caused by each of these species (or groups of species Bpc/Bcc) what is similar and what is different including a figure to orient the readers before going into an in-depth discussion of virulence factors and the development of many as vaccine candidates. For example, are the virulence strategies of Bpc and Bcc similar? Even a figure with each bacteria demonstrating where the location of cross-reactive antigens are would be useful.

Again, the discussion talks about the extracellular phase of infection, I think a description of the basic virulence of each group would be helpful in the introduction and describe if the goal of a vaccine is sterilizing immunity or if the goal is different.

Response: Thank you for the feedback. We have added a couple paragraphs to the introduction to try and address this concern. We tried to keep this information somewhat general since we go into many of the specific virulence mechanisms throughout the review. In responding to this comment, we primarily focused on describing pathogenesis of the Bpc pathogens, not Bcc. This is because Bcc pathogenesis virulence mechanisms are far less defined than those of the Bpc and there are also many subtle differences between the different species of the Bcc that makes a discussion of the topic difficult. Because the primary focus of this review is providing a comprehensive overview of Bpc vaccines, we feel that it is sufficient to discuss only their disease pathogenesis in detail, especially given how long the review already is.

2.) As the entire review deals with glycoconjugate and subunit vaccines, a brief description of how these elicit immune responses would be beneficial. Also, a table of which adjuvants seem to work best in subunit formulations may be beneficial to the readers.

Response: Regarding the first part of your comment, we have tried to tie our discussion of disease pathogenesis (per your first suggestion) to correlates of protection that need to be elicited by a Bpc vaccine. Specifically, we discuss how antibodies are protective during the extracellular phase of the infection while T cells are primarily mediating protection during intracellular stages of infection. Given how long the review already is we did not think it would be prudent to give a full review of all immunological aspects, so we tried to keep our language relatively simple and to the point in this section.

Regarding including a table of what adjuvants have been used, we had originally attempted to make sense of this and identify trends in the adjuvants used by different studies but were ultimately unsuccessful. Most of the subunit vaccine studies that we reviewed did not include T cell recall data or immunoglobulin subtyping data which would have helped to make sense of how these different adjuvants are promoting or impairing vaccine efficacy. As such, we could glean relatively little information about what adjuvants are responsible for poor vaccine efficacy and very few studies have specifically set out to compare different adjuvants side-by-side.

3) If the goal was to make a cross-protective glycoconjugate vaccine would you have to include Bcc polysaccharides?

Response: This is an excellent question. The short answer is that it is highly unlikely that any polysaccharide antigens will have applicability as cross-reactive antigens due to the high level of diversity in surface polysaccharide composition among members of the cepacia complex. It is possible that a combination of pseudomallei complex- and cepacia complex-derived polysaccharides could be used in the same vaccine formulation if humoral responses to such antigens are desired or required. However, since Cloutier et al published an extremely comprehensive review of Burkholderia polysaccharides and their use in pseudomallei complex vaccines (Nat. Prod. Rep., 2018, 35, 1251-1293), we decided not to include an extensive review here of polysaccharides. Instead, we have included a significantly condensed version on polysaccharide antigens that we hope highlights the most important information.

4) I think the title is somewhat misleading because a lot of the text is about vaccine candidates that have no potential to elicit cross-protection. With this in mind, this review could be drastically shortened. It may be more pertinent to describe the best vaccine candidates for Bcp and what types of immune responses produce protection. Then specifically address proteins with potential cross-protection and their ability to provide similar protective responses.

Response: We thank you for the suggestion. This is something we have gone back-and-forth on during the writing process. When we first started conceptualizing this review, we originally wanted to provide a comprehensive review of what antigens have been tested in Burkholderia pseudomallei complex vaccines, their efficacy, and potential avenues of future research, as this topic has not been comprehensively reviewed in recent years. However, as the writing progressed, we became increasingly interested in the concept of a pan-Burkholderia vaccine and started looking into which, if any, of these antigens could potentially serve in such a vaccine. We were pleasantly surprised to find that a handful of efficacious antigens were broadly conserved across the cepacia complex, and we wanted to share this information with other researchers. Because we feel that both focuses are beneficial to Burkholderia vaccine researchers, we are of the opinion that it would be detrimental to the review to remove the sections on poorly conserved antigens or antigens that failed to elicit protection, which would only really leave the three sections on flagellar proteins, lipoproteins, and outer membrane beta barrel proteins.

5) The conclusions discuss outdated models of infection, and the introduction could benefit from a description of which models the authors believe are the most appropriate for studying vaccination.

Response: We added additional information to convey our opinion that the antigens that were previously studied in i.p. challenge models should be revisited using the currently favored models of respiratory acquired melioidosis and glanders. We have also added a couple sentences explaining the field’s shift from i.p. to respiratory challenge models.

6) Lines 627-628 indicate Hcp1 is most likely a T cell antigen. However, Lines 630-632 demonstrate that human sera respond to Hcp1. Can the authors provide a comment on this?

Response: We neglected to properly define the term “T cell antigen” as it is used in the review, so the confusion about these statements is understandable. We attempted to provide additional explanation of these terms in the new introduction paragraph that we added in response to your second critique/suggestion. In brief, when we use the term “T cell antigen” in the review, we do not mean to say that antibodies cannot be elicited to these antigens. On the contrary, antibodies often can be elicited to such “T cell antigens;” however, because these antigens are either located beneath the outer membrane or are secreted into the intracellular environment of the host cell, they are generally not thought to engage antibodies during a productive infection. As such, antibodies against such antigens are not predicted to contribute to protection. Any protection conferred by these antigens would instead be expected to be due to T cell responses, hence their designation as “T cell antigens.”

Reviewer 2 Report

Comments and Suggestions for Authors

Dear Authors and Editor,

Thank you for the interesting read. This paper is a very thorough investigation of the field of Burkholderia vaccinology to the aspiration of understand the potential for vaccines developed for B. pseudomallei/mallei protecting against cepacia syndrome. The authors go into considerable detail regarding the all of the published vaccine antigens and use some simple bioinformatics methods to consider if these vaccine strategies may be beneficial to prevent infection from the cenocepacia group of pathogens. These opportunist pathogens that are most profoundly dangerous in people with cystic fibrosis.

The article is well-written and well researched and covers an impressive breadth of the field.

I think that the authors have failed to consider one key question. That is, can the types of adaptive immunity generated by vaccination be beneficial against the types of infection caused by this group of pathogens. I am no expert in cepacia syndrome however I understand that the incorrect function of mucus in the lung allows for the establishment of biofilms by pathogens such as B. cepacia. These bacteria are protected against antibiotics and plausibly the adaptive immune response. I wonder if any vaccine can protect against this type of infection. A cursory look reveals that attempts to prevent Pseudomonas aeruginosa infection, in cystic fibrosis patients by vaccination have been disappointing (https://www.ncbi.nlm.nih.gov/pmc/articles/PMC7055511/#:~:text=The%20large%20trial%20also%20reported,in%20patients%20with%20cystic%20fibrosis.). I would recommend that this article would benefit in a frank discussion about the difficulties in preventing these types of infection by vaccination and how these might be overcome. For example, might a B. pseudomallei vaccine with cross reactivity for B. cepacia antigens be improved if it could be reformulated or delivered by an alternative route to elicit a greater IgA response? Would this even be beneficial? Perhaps this is something that the authors did not consider as important to the publication, which they wanted concentrated of shared antigens. However, I think that this consideration is essential for exploitation and like to see even some minimal acknowledgement that this could be a complication.

Best regards,

Reviewer.

Author Response

Response to reviewers of manuscript ID: vaccines-2850735: “Burkholderia pseudomallei Complex Subunit and Glycoconjugate Vaccines and Their Potential to Elicit Cross-Protection to Burkholderia cepacia Complex”.

We greatly appreciate your editorial suggestions to ensure that our manuscript is in accordance with Vaccines and MDPI guidelines. Additionally, we appreciated the reviewers’ suggestions and criticism to our manuscript. We have enclosed a revised manuscript for your consideration that has been modified in line with all the reviewers’ comments. Find appended below the details of the changes made in response to these comments.

Reviewer 2.

1) I think that the authors have failed to consider one key question. That is, can the types of adaptive immunity generated by vaccination be beneficial against the types of infection caused by this group of pathogens. I am no expert in cepacia syndrome however I understand that the incorrect function of mucus in the lung allows for the establishment of biofilms by pathogens such as B. cepacia. These bacteria are protected against antibiotics and plausibly the adaptive immune response. I wonder if any vaccine can protect against this type of infection. A cursory look reveals that attempts to prevent Pseudomonas aeruginosa infection, in cystic fibrosis patients by vaccination have been disappointing (https://www.ncbi.nlm.nih.gov/pmc/articles/PMC7055511/). I would recommend that this article would benefit in a frank discussion about the difficulties in preventing these types of infection by vaccination and how these might be overcome. For example, might a B. pseudomallei vaccine with cross reactivity for B. cepacia antigens be improved if it could be reformulated or delivered by an alternative route to elicit a greater IgA response? Would this even be beneficial? Perhaps this is something that the authors did not consider as important to the publication, which they wanted concentrated of shared antigens. However, I think that this consideration is essential for exploitation and like to see even some minimal acknowledgement that this could be a complication.

Response: Thank you for pointing this out. This is indeed a topic that we neglected to mention that certainly warrants discussion. We have attempted to review the literature on vaccines against bacteria that colonize the CF lung and have included a brief summary in the Conclusion/Future Directions section.  Whether or not protective immunity can be achieved in the unique environment of the CF lung is certainly a major concern of this approach and one that does not yet appear to have any clear-cut answers. Frustratingly, despite decades of research it still isn’t well understood how the CF lung environment impairs different arms of the immune system, so it cannot be said definitively whether vaccinating against CF-associated pathogens is possible or not. That said, there are at least a couple examples of human CF P. aeruginosa vaccine clinical trials that showed a modest ability of the vaccine to protect patients from disease. We believe that future attempts to develop vaccines for CF patients will make use of modern approaches to stimulating mucosal immunity, which appears to be an oversight of previous CF vaccine clinical trials.

Reviewer 3 Report

Comments and Suggestions for Authors

This paper focus on the beliefs and theories on vaccination and especially the adherence and investigating some factors shaping public health. It is important review study and needs some minor rervision, s specified in attached file.

Author Response

Response to reviewers of manuscript ID: vaccines-2850735: “Burkholderia pseudomallei Complex Subunit and Glycoconjugate Vaccines and Their Potential to Elicit Cross-Protection to Burkholderia cepacia Complex”.

We greatly appreciate your editorial suggestions to ensure that our manuscript is in accordance with Vaccines and MDPI guidelines. Additionally, we appreciated the reviewers’ suggestions and criticism to our manuscript. We have enclosed a revised manuscript for your consideration that has been modified in line with all the reviewers’ comments. Find appended below the details of the changes made in response to these comments.

Reviewer 3.

This paper focus on the beliefs and theories on vaccination and especially the adherence and investigating some factors shaping public health. It is important review study and needs some minor revision is specified in attached file.

Response: Thank you for your minor revisions, they have been incorporated in the revised version of the manuscript.